# Diversity-Preserved Distribution Matching Distillation for Fast Visual Synthesis

**Tianhe Wu** [* 1 3]  **Ruibin Li** [* 2]  **Lei Zhang** [† 2 3]  **Kede Ma** [† 1]

https://github.com/Multimedia-Analytics-Laboratory/dpdmd

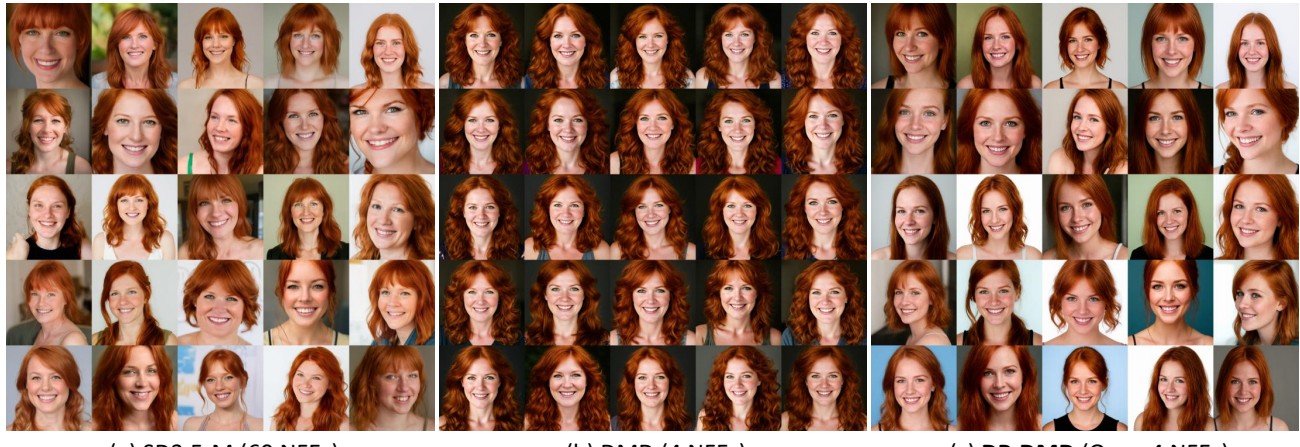

(a) SD3.5-M (60 NFEs)    (b) DMD (4 NFEs)    (c) **DP-DMD** (Ours, 4 NFEs)

Figure 1. **DP-DMD preserves sample diversity** while maintaining competitive perceptual quality. All results are generated under identical text conditioning ("*A smiling woman with red hair, green eyes, and dimples.*") with different random initial noise. **(a)** SD3.5-M (Esser et al., 2024), sampled with 30 steps (*i.e.*, 60 NFEs), serves as the teacher model. **(b)** DMD (Yin et al., 2024a) and **(c)** DP-DMD are step-distilled student models, both evaluated using only 4 NFEs.

## Abstract

Distribution matching distillation (DMD) facilitates few-step image generation by aligning a distilled student with a reference multi-step teacher. In practice, however, optimizing DMD can reduce sample diversity in few-step synthesis, and existing remedies typically rely on perceptual or adversarial regularization, leading to stability and scalability challenges during training. Here, we describe diversity-preserved DMD (**DP-DMD**), a role-separated distillation method inspired by the complementary roles of early and late denoising steps. Specifically, the first distillation step is trained with a teacher-derived target-prediction objective (*e.g.*, $v$-prediction) to preserve sample diversity, while the remaining steps are optimized with the standard DMD loss to refine perceptual quality. DP-DMD, with *no perceptual or adversarial regularization, no additional modules, and no teacher-generated reference samples*, preserves sample diversity while maintaining competitive visual quality under few-step sampling, providing a simple and stable alternative to other DMD variants.

## 1. Introduction

Recent years have witnessed rapid progress in generative modeling, particularly with diffusion (Song et al., 2020) and flow-based models (Lipman et al., 2022). These approaches have opened the door to high-quality image and video synthesis (Esser et al., 2024; Wan et al., 2025), supported by advances in large foundation models (Rombach et al., 2022; Labs et al., 2025) and modern optimization techniques (Sutton et al., 1999; Liu et al., 2025b). Despite their

---
[*]Equal contribution. [1]City University of Hong Kong, [2]The Hong Kong Polytechnic University, [3]OPPO Research Institute. Correspondence to: Lei Zhang <cslzhang@comp.polyu.edu.hk> and Kede Ma <kede.ma@cityu.edu.hk>.

*Proceedings of the 43[rd] International Conference on Machine Learning*, Seoul, South Korea. PMLR 306, 2026. Copyright 2026 by the author(s).

strong performance, these models typically require solving multi-step dynamical systems during sampling, leading to substantial inference latency and computational cost due to the large number of function evaluations (NFEs).

To reduce sampling cost, prior distillation methods have sought to "compress" a reference multi-step teacher into a few-step student. Early efforts mainly follow a trajectory-based approach (Liu et al., 2023b; Yan et al., 2024), where the student is trained to approximate the teacher's denoising trajectories using fewer steps. More recently, a complementary line of work has shifted from trajectory imitation to *distribution matching*, directly aligning the student distribution with that of the teacher (Sauer et al., 2024b; Yin et al., 2024b;a; Liu et al., 2025a). Among these, distribution matching distillation (DMD, Yin et al. 2024a;b) has emerged as a particularly effective objective, offering a strong balance between fast sampling and high visual quality.

Nevertheless, DMD suffers from an important limitation: it often reduces sample diversity in few-step generation. As shown in Figure 1, student models optimized with the DMD loss can produce visually plausible but substantially less diverse samples than their multi-step teachers. This behavior is consistent with the mode-seeking tendency of the reverse-KL-style objective underlying DMD, which may weaken coverage of the teacher distribution, especially for lower-probability modes. In practice, this issue becomes more pronounced when aggressive step reduction leaves the student with limited capacity to capture the full diversity of the teacher distribution.

Existing attempts to mitigate this issue typically augment DMD with perceptual or adversarial regularization (Yin et al., 2024a;b; Chadebec et al., 2025; Lu et al., 2025). These regularizers encourage broader coverage of the teacher distribution, often by leveraging additional teacher-generated samples as implicit supervision. However, such designs also introduce notable drawbacks: perceptual losses increase memory and computation overhead[1], while adversarial losses often make training less stable and more difficult to scale. These issues motivate a simpler and more stable way to preserve diversity during DMD.

In this paper, we describe **diversity-preserved DMD (DP-DMD)**, a role-separated distillation method motivated by a simple observation about the denoising process: early denoising steps mainly determine global image structure and thus play a dominant role in sample diversity, whereas later steps primarily refine local details and perceptual quality (see Figure A in the Appendix). Based on this asymmetry, DP-DMD adopts different optimization objectives to

---

[1] Widely used perceptual losses, such as LPIPS (Zhang et al., 2018) and DISTS (Ding et al., 2020), incur non-trivial GPU memory usage and computational cost, especially when applied to high-resolution imagery.

guide distinct distillation steps. The first step is trained with a teacher-derived target-prediction objective (*e.g.*, $v$-prediction) to preserve sample diversity, while the remaining steps are optimized using the standard DMD loss to improve image quality. To enforce this functional separation, gradients from the DMD loss are stopped (*i.e.*, detached) at the first step, preventing the mode-seeking reverse-KL objective from overriding the diversity-preserving supervision.

A key advantage of DP-DMD is its *simplicity*. The method requires *no perceptual or adversarial regularization, no additional modules, or teacher-generated reference images*. All computations are performed in latent space, resulting in a simple, stable, and memory-efficient training pipeline. Despite its minimalist design, DP-DMD consistently improves the diversity-quality trade-off in few-step text-to-image generation across diffusion-based and flow-based backbones, supported by subjective verification.

## 2. Related Work

Research on accelerating diffusion and flow-based generative models can be broadly grouped into two families: *trajectory-based distillation*, which compresses the teacher's denoising paths, and *distribution-level matching*, which aligns the student distribution with that of the teacher.

**Trajectory-based distillation.** Trajectory-based distillation aims to compress the teacher's denoising or transport trajectories into a student that requires substantially fewer inference steps. Representative methods include consistency models (Song et al., 2023) and their improved variants (Song & Dhariwal, 2023; Wang et al., 2024; Lu & Song, 2024), which typically select anchor points along the teacher trajectories and train the student to reproduce the corresponding transitions. More recently, MeanFlow (Geng et al., 2025) further investigates trajectory compression by matching average diffusion velocities. Although trajectory-based methods have shown strong acceleration performance, their effectiveness often degrades in extremely low-step regimes, particularly for large-scale pre-trained image and video generators, where faithfully preserving the teacher's full trajectories becomes increasingly challenging (Zheng et al., 2025). This limitation has motivated alternative approaches that relax strict trajectory imitation and instead pursue distribution-level matching to the teacher.

**Distribution-level matching.** A complementary line of work distills generative models by aligning the student distribution with that of a pre-trained teacher model. One branch adopts adversarial regularization, either by introducing dedicated discriminators or by repurposing diffusion features to construct generative adversarial network (GAN)-like objectives (Sauer et al., 2024a;b; Lin et al., 2024; Zhou et al., 2024a; Chadebec et al., 2025). Although these meth-

ods tend to improve visual sharpness and reduce the number of sampling steps, the added adversarial components often make optimization much more sensitive and less scalable.

Another branch is inspired by distribution-matching ideas originally developed in text-to-3D generation (Poole et al., 2022; Wang et al., 2023a), where optimization is driven by discrepancies between synthesized samples and diffusion model guidance. DMD transfers this principle to image synthesis distillation and achieves a strong balance between efficiency and sample quality (Yin et al., 2024b). Subsequent extensions further improve performance through refined objectives (Zhou et al., 2024b; Luo et al., 2025), guidance strategies (Liu et al., 2025a), or auxiliary training mechanisms (Yin et al., 2024a; Zheng et al., 2025).

Nevertheless, DMD-based methods exhibit a reverse-KL-like mode-seeking tendency, which can reduce support coverage and harm sample diversity under aggressive step reduction. Existing remedies often introduce perceptual or adversarial regularization (Yin et al., 2024b;a; Chadebec et al., 2025), which may increase complexity, destabilize training, and hinder scalability. In contrast, DP-DMD improves sample diversity through step-wise loss design, preserving diversity in the first distillation step and using later steps for standard DMD-based visual quality refinement.

# 3. Preliminaries

We briefly review flow matching and DMD, which together form the technical foundation of the proposed DP-DMD.

## 3.1. Flow Matching

Diffusion and flow-based generative models can be formulated in continuous-time ordinary differential equations (Chen et al., 2018; Song et al., 2020). Let $x \sim p_{\text{data}}$ denote a clean data sample and $\epsilon \sim p_{\text{noise}}$ denote a noise sample, where typically $p_{\text{noise}} \triangleq \mathcal{N}(\mathbf{0}, \mathbf{I})$. A continuous path between data and noise is constructed as $z_t = a_t x + b_t \epsilon$, for $t \in [0, 1]$, where $a_t$ and $b_t$ are predefined schedules. Following prior work (Liu et al., 2023b; Esser et al., 2024), we adopt the linear schedule:

$$z_t = (1 - t)x + t\epsilon. \qquad (1)$$

Under this construction, $z_t$ evolves from the data distribution at $t = 0$ to the noise distribution at $t = 1$.

The corresponding velocity field is given by the time derivative of $z_t$. For the linear path in Equation (1), we have

$$v_t = \frac{dz_t}{dt} = \epsilon - x. \qquad (2)$$

Flow matching learns a neural velocity field $v_\theta(z_t, t)$ by regressing it to the target velocity along this path (Lipman

et al., 2022; Geng et al., 2025):

$$\ell_{\text{FM}}(\theta) = \mathbb{E}_{t, x, \epsilon}\left[\left\|v_\theta(z_t, t) - (\epsilon - x)\right\|^2\right]. \qquad (3)$$

At inference time, generation starts from $z_1 \sim p_{\text{noise}}$ and integrates the learned ODE backward from $t = 1$ to $t = 0$: $\frac{dz_t}{dt} = v_\theta(z_t, t)$. The resulting terminal state defines the generated sample: $x_\theta = z_1 + \int_1^0 v_\theta(z_t, t)\, dt$.

## 3.2. DMD Loss

Given a pre-trained multi-step teacher and a few-step student, DMD seeks to align the student distribution $p_{\text{stu}}$ with the teacher distribution $p_{\text{tea}}$. Let $x_\theta = g_\theta(\epsilon)$ denote a sample generated by the student, where $\epsilon \sim p_{\text{noise}}$ is a noise variable and thus $x_\theta \sim p_{\text{stu}}$. The DMD loss is defined as

$$\ell_{\text{DMD}}(\theta) \triangleq D_{\text{KL}}\big(p_{\text{stu}}(x_\theta) \,\|\, p_{\text{tea}}(x_\theta)\big). \qquad (4)$$

Directly evaluating this divergence in high-dimensional image space is intractable due to *implicit density representations* and *support mismatch* between two distributions (Poole et al., 2023; Wang et al., 2023a). Instead, DMD computes its gradient in a shared perturbed space. Specifically, the student sample $x_\theta$ is diffused according to Equation (1) to obtain $z_t$, which brings the teacher and student distributions into overlapping support (Yin et al., 2024b). Under this assumption, the gradient of $\ell_{\text{DMD}}(\theta)$ with respect to the student parameters $\theta$ can be written as

$$\nabla_\theta \ell_{\text{DMD}}(\theta) = \mathbb{E}_{x_\theta}\left[\left(\nabla_\theta x_\theta\right)^\mathsf{T}\big(s_{\text{stu}}(z_t) - s_{\text{tea}}(z_t)\big)\right], \quad (5)$$

where $s_{\text{stu}} = \nabla_{z_t} \log p_{\text{stu}}(z_t)$ and $s_{\text{tea}} = \nabla_{z_t} \log p_{\text{tea}}(z_t)$ are the score functions of the student- and teacher-induced distributions in the perturbed space, respectively.

In practice, the teacher-side score can be obtained from the pre-trained multi-step teacher. By contrast, the score of the student-induced distribution is not explicitly parameterized by the distilled generator and is generally intractable to compute in closed form. Therefore, following prior DMD methods (Yin et al., 2024a), we approximate it using an auxiliary "fake" model[2].

# 4. Proposed Method: DP-DMD

We present DP-DMD, a role-separated distillation method for few-step visual synthesis, motivated by the observation that different portions of the denoising trajectory contribute differently to the final sample. The training pipeline is shown in Figure 2.

---

[2]The auxiliary model is initialized in the same way as the teacher with a shared architecture.

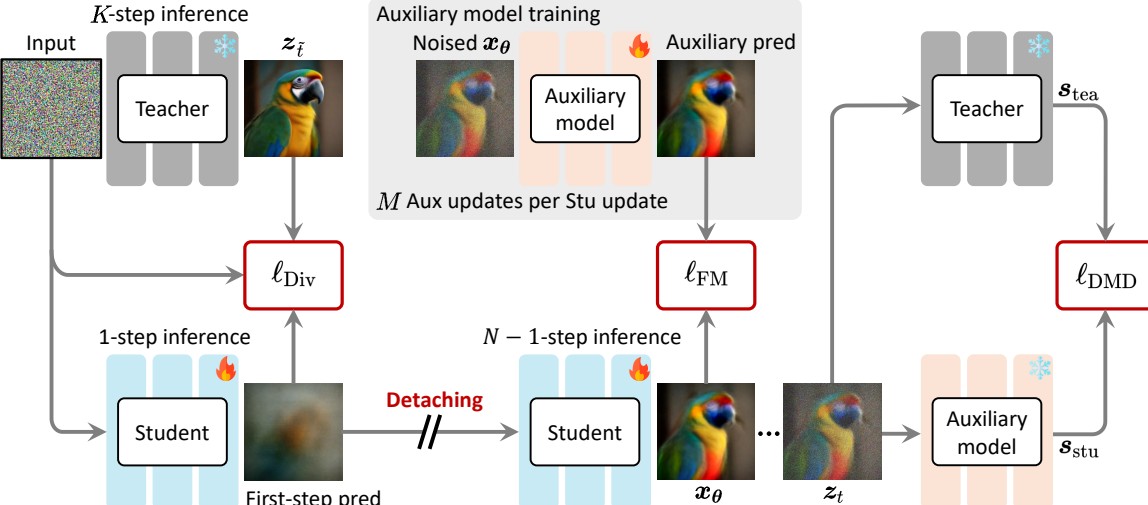

Figure 2. **Training pipeline of DP-DMD**. The student's first denoising step is supervised by a teacher-derived intermediate state through the target-prediction loss $\ell_{\text{Div}}$. Gradients are detached after this step to prevent the DMD loss from overriding the early diversity signal. The remaining $N-1$ steps are trained with the standard DMD loss, where teacher and auxiliary model scores provide distribution-matching guidance for perceptual quality refinement. This role-separated design preserves sample diversity while maintaining high-quality few-step generation.

## 4.1. Roles of Early and Late Denoising Steps

Diffusion and flow-based generative models exhibit a salient stage-wise denoising behavior (Wang & Vastola, 2023; Liu et al., 2023a; He et al., 2025), which is central to our design.

**Early-step sample diversity preservation.** Early denoising steps operate at high noise levels and are primarily responsible for establishing the coarse geometric and semantic structure of a sample, including overall composition, coarse geometry, and object presence. As shown in Figure A of the Appendix, variation introduced at this stage is carried forward by the subsequent trajectory and remains visible in the final output. These early decisions therefore play a dominant role in determining sample diversity.

**Late-step perceptual quality refinement.** In contrast, later denoising steps operate at lower noise levels and mainly refine local visual details, such as texture, colors, contours, and fine appearance. Since the global layout has typically been determined by the previous stage, these steps have a weaker effect on sample diversity and contribute more directly to perceptual quality.

This asymmetry suggests that applying the same distillation loss to all steps, as in standard DMD (Yin et al., 2024a;b), is not ideal for preserving sample diversity under aggressive step reduction. DP-DMD addresses this issue by explicitly separating the roles of the distillation steps: the first step is trained to preserve diverse global structure, and the subsequent steps are trained to improve visual quality.

## 4.2. Details of DP-DMD

Consider a distilled student model with $N$ inference steps. DP-DMD decomposes the training objective according to the role of each step.

**Teacher-derived diversity supervision.** We first construct a supervision target for the diversity-preserving step. Starting from the same initial noise $\boldsymbol{\epsilon}$, we run the multi-step teacher for $K$ denoising steps and denote the resulting intermediate latent by $\boldsymbol{z}_{\tilde{t}}$, where $\tilde{t}$ is the corresponding continuous-time index. The anchor step $K$ determines which teacher state is used to guide the student's first step and therefore controls the strength of the diversity-preserving signal.

Under the linear flow path in Equation (1), the teacher-derived velocity that transports the initial noise toward $\boldsymbol{z}_{\tilde{t}}$ for the student's first denoising step is

$$\tilde{\boldsymbol{v}} = \frac{\boldsymbol{\epsilon} - \boldsymbol{z}_{\tilde{t}}}{1 - \tilde{t}}. \tag{6}$$

The student predicts a velocity field $\boldsymbol{v}_{\boldsymbol{\theta}}(\boldsymbol{\epsilon}, 1)$ from the initial noise. We supervise this prediction using the diversity loss:

$$\ell_{\text{Div}}(\boldsymbol{\theta}) = \mathbb{E}_{\boldsymbol{\epsilon}}\left[\left\|\boldsymbol{v}_{\boldsymbol{\theta}}(\boldsymbol{\epsilon}, 1) - \tilde{\boldsymbol{v}}\right\|^2\right]. \tag{7}$$

By aligning the first-step prediction with a teacher-derived intermediate state, this loss encourages the student to retain diverse global structure encoded by the teacher's early denoising trajectories. Importantly, this supervision is applied only at the first step, where the teacher and student share the same input noise distribution.

**Algorithm 1** DP-DMD training for flow-based models

```
# x: training data batch
# K: diversity anchor step index
# t_tilde: anchor diffusion timestep
# N: number of student sampling steps
# lambda_div: diversity loss weight

eps = randn_like(x)
z_tilde = rollout_teacher(eps, K)

# diversity supervision
v_tilde = (eps - z_tilde) / (1 - t_tilde)
v1 = v_stu(eps, 1)
loss_div = l2_loss(v1 - v_tilde)

# detach after first step
z1 = stopgrad(rollout_student(eps, 1))

# quality supervision
x_theta = rollout_student(z1, N - 1)
loss_dmd = dmd_loss(x_theta)

loss = loss_dmd + lambda_div * loss_div
```

**DMD-based quality supervision.** After the first denoising step, the student is rolled out for the remaining $N - 1$ steps to obtain the final sample $x_\theta$. To maintain a clean separation between diversity preservation and quality refinement, we detach the output of the first step from the computational graph before applying the DMD loss. Consequently, gradients from DMD cannot be propagated back to the diversity-preserving step. The full DP-DMD loss is

$$\ell(\boldsymbol{\theta}) = \ell_{\text{DMD}}(\boldsymbol{\theta}) + \lambda\, \ell_{\text{Div}}(\boldsymbol{\theta}), \quad (8)$$

where $\lambda$ controls the trade-off between the two terms. DP-DMD therefore requires no perceptual loss, adversarial discriminator, additional modules, or stored teacher-generated reference images. Algorithm 1 summarizes one training iteration for flow-based models.

# 5. Experiments

In this section, we first describe the training and evaluation protocol. We then compare DP-DMD with regularization-based DMD variants and existing few-step methods, focusing on how different objectives affect the diversity-quality trade-off. Finally, we conduct ablation studies to analyze the design choices responsible for the observed behavior, along with additional prompt-following evaluation.

## 5.1. Experimental Setups

**Training.** We use SD3.5-Medium (Esser et al., 2024) and SDXL (Podell et al., 2023) as representative flow-based and diffusion-based text-to-image teachers. Unless otherwise specified, all models are distilled at $1,024 \times 1,024$ resolution. The teacher classifier-free guidance (CFG) scale is fixed to $3.5$ for SD3.5-M and $8.0$ for SDXL. The student generator and the auxiliary "fake" model used for DMD

score estimation are optimized with AdamW (Loshchilov & Hutter, 2017) using a learning rate of $10^{-5}$. We set the diversity weight to $\lambda = 5 \times 10^{-2}$, update the auxiliary model every $M = 5$ student updates, and use $K = 5$ as the default diversity anchor. Distillation is performed on DiffusionDB prompts (Wang et al., 2023b) for $6 \times 10^3$ iterations, with a per-GPU batch size of $4$ on $8$ NVIDIA A800 GPUs. When comparing DMD variants, we keep the training budget, CFG scale, and inference cost matched.

**Evaluation.** The evaluation focuses on the central goal of this work: improving sample diversity without sacrificing perceptual quality (or text-image preference). For each prompt, we generate multiple samples from different initial noise and compute feature-space diversity as

$$\text{Diversity} = 1 - \frac{2}{R(R-1)} \sum_{i,j} \cos\left(\boldsymbol{x}_{\boldsymbol{\theta}}^{(i)}, \boldsymbol{x}_{\boldsymbol{\theta}}^{(j)}\right), \quad (9)$$

where $R$ denotes the number of initial noise samples per prompt, and we set $R = 9$ in all diversity evaluations. We report diversity using DINOv3-ViT-Large (DINO, Siméoni et al. 2025) and CLIP-ViT-Large (CLIP, Radford et al. 2021). DINO emphasizes structural variation, while CLIP is more sensitive to semantic variation under the same text condition. Perceptual quality is measured with VisualQuality-R1 (VQ-R1, Wu et al. 2025) and MANIQA (MIQA, Yang et al. 2022); preference alignment is measured with ImageReward (ImgR, Xu et al. 2023) and PickScore (PicS, Kirstain et al. 2023). We therefore interpret results jointly rather than treating any single metric as decisive.

## 5.2. Main Results

**Controlled comparison of diversity supervision.** We compare DP-DMD with two representative strategies for mitigating diversity degradation in DMD: perceptual regularization with LPIPS and adversarial regularization with a discriminator. All variants in Table 1 use the same backbone, training data, training budget, CFG scale, and NFEs.

Vanilla DMD is a strong few-step baseline because it directly aligns the final student distribution with the teacher through score differences in a perturbed space. However, this objective is reverse-KL-like and therefore tends to be mode-seeking: under a small NFEs, the student can reduce the loss by mapping different noise inputs to a narrower set of high-density teacher modes. This explains its strong quality but reduced sample diversity.

DMD-LPIPS partially alleviates this issue by adding a perceptual feature constraint. Since LPIPS measures sample-level perceptual similarity rather than distributional support coverage, its gradients are only indirectly related to preserving the branching structure of the denoising trajectory. As a result, while DMD-LPIPS can increase apparent diversity

Table 1. **Controlled comparison of DMD variants** on Pick-a-Pic ([Kirstain et al., 2023](#)) and COCO-10K 2014 ([Lin et al., 2014](#)). All variants are evaluated with the same backbone and inference budget. The best and second-best results are highlighted in **bold** and underline, respectively.

| Method | Image-Free | NFEs | Pick-a-Pic | | | | | | COCO-10K 2014 | | | | | |
|---|---|---|---|---|---|---|---|---|---|---|---|---|---|---|
| | | | Diversity | | Quality | | Preference | | Diversity | | Quality | | Preference | |
| | | | DINO↑ | CLIP↑ | VQ-R1↑ | MIQA↑ | ImgR↑ | PicS↑ | DINO↑ | CLIP↑ | VQ-R1↑ | MIQA↑ | ImgR↑ | PicS↑ |
| *SD3.5-M* (CFG=3.5) | | | | | | | | | | | | | | |
| Base Model | - | 60 | 0.240 | 0.221 | 4.657 | 1.020 | 1.007 | 21.80 | 0.288 | 0.204 | 4.636 | 1.043 | 0.910 | 22.31 |
| DMD | ✓ | 4 | 0.137 | 0.133 | **4.649** | 1.016 | **1.189** | 21.75 | 0.210 | 0.154 | **4.690** | **1.060** | **1.053** | 22.40 |
| DMD-LPIPS | ✗ | 4 | 0.169 | 0.169 | 4.598 | 1.005 | 1.063 | 21.62 | 0.204 | 0.168 | 4.599 | 1.012 | 0.949 | 22.29 |
| DMD-GAN | ✗ | 4 | **0.183** | 0.162 | 4.525 | 0.984 | 1.033 | 21.63 | 0.214 | 0.174 | 4.584 | 0.983 | 0.751 | 22.02 |
| **DP-DMD** | ✓ | 4 | 0.179 | **0.182** | 4.646 | **1.017** | 1.142 | **21.76** | **0.250** | **0.182** | 4.689 | 1.032 | 0.988 | **22.41** |
| *SDXL* (CFG=8.0) | | | | | | | | | | | | | | |
| Base Model | - | 100 | 0.219 | 0.204 | 4.675 | 1.033 | 1.016 | 21.96 | 0.269 | 0.219 | 4.637 | 1.056 | 0.820 | 22.54 |
| DMD | ✓ | 4 | 0.109 | 0.133 | **4.667** | 0.971 | 0.951 | 21.68 | 0.139 | 0.143 | 4.643 | 0.982 | 0.712 | 22.21 |
| DMD-LPIPS | ✗ | 4 | 0.136 | 0.139 | 4.610 | 0.976 | 0.883 | 21.74 | 0.181 | 0.137 | 4.723 | 0.984 | 0.729 | 22.38 |
| DMD-GAN | ✗ | 4 | 0.126 | 0.124 | 4.624 | **1.019** | **1.036** | **21.80** | 0.157 | 0.117 | **4.789** | 1.030 | 0.801 | **22.63** |
| **DP-DMD** | ✓ | 4 | **0.173** | **0.161** | 4.591 | 0.954 | 1.011 | 21.75 | **0.204** | **0.157** | 4.765 | **1.041** | **0.835** | 22.45 |

Table 2. **Practical comparison** of open-source few-step methods.

| Method | Image-Free | NFEs | Pick-a-Pic | | | | | | COCO-10K 2014 | | | | | |
|---|---|---|---|---|---|---|---|---|---|---|---|---|---|---|
| | | | Diversity | | Quality | | Preference | | Diversity | | Quality | | Preference | |
| | | | DINO↑ | CLIP↑ | VQ-R1↑ | MIQA↑ | ImgR↑ | PicS↑ | DINO↑ | CLIP↑ | VQ-R1↑ | MIQA↑ | ImgR↑ | PicS↑ |
| Base Model | - | 60 | 0.230 | 0.205 | 4.665 | 1.042 | 1.067 | 21.46 | 0.278 | 0.202 | 4.757 | 1.065 | 1.014 | 22.50 |
| Hyper-SD | ✗ | 8 | **0.234** | **0.225** | 4.268 | 0.917 | 0.808 | 20.77 | **0.297** | **0.236** | 3.929 | 0.924 | 0.661 | 21.56 |
| Flash | ✗ | 4 | 0.184 | 0.172 | 4.625 | **1.023** | 1.014 | **21.62** | 0.229 | 0.184 | 4.517 | 0.998 | 0.878 | **22.38** |
| TDM | ✓ | 4 | 0.148 | 0.167 | **4.675** | 1.013 | **1.134** | 21.21 | 0.196 | 0.172 | 4.617 | 1.046 | 0.998 | 21.49 |
| **DP-DMD** | ✓ | 4 | 0.162 | 0.181 | 4.673 | 1.001 | 1.128 | 21.15 | 0.197 | 0.174 | **4.672** | **1.048** | **1.034** | 22.29 |

in some cases, it does not reliably preserve the underlying branching structure needed for support coverage.

DMD-GAN supplies an adversarial signal that can push samples toward the natural image manifold and sometimes broaden visual variation. Nevertheless, the discriminator introduces a minimax optimization problem, making training less stable, and encouraging artifact amplification and dataset-specific shortcuts. The visual results in Figure 3 show that apparent diversity gains may coincide with degraded quality, which in turn make feature-space diversity scores less reliable.

DP-DMD avoids these drawbacks by changing the allocation of distillation losses rather than adding external regularizers. This simple design improves diversity over vanilla DMD while maintaining competitive quality and preference, at essentially no cost. The subjective user study in Section D of the Appendix further supports these findings.

**Practical comparison with few-step methods.** We further compare DP-DMD with representative open-source few-step methods in Table 2. This comparison is not strictly controlled because the methods may differ in training data,

teacher configuration, guidance scale, optimization budget, and implementation details. It nevertheless helps assess whether the proposed design remains practical beyond the matched DMD-variant setting.

Hyper-SD ([Ren et al., 2024](#)) relies on trajectory-style compression, which preserves variation by referencing intermediate teacher states but may leave limited capacity for final-detail refinement under very small step budgets. Flash Diffusion ([Chadebec et al., 2025](#)) uses adversarial distillation, which improves sharpness but inherits the tuning sensitivity. TDM ([Luo et al., 2025](#)) matches trajectory distributions and remains image-free, but it does not explicitly reserve the earliest student step for diversity preservation, so global branching and final refinement must still be balanced within the same compressed rollout. By contrast, DP-DMD uses a simpler, image-free decomposition, maintaining competitive sample diversity, perceptual quality, and prompt consistency at low inference cost (see also Figure 4).

### 5.3. Ablation Studies

All ablation studies are conducted on Pick-a-Pic ([Kirstain et al., 2023](#)) under the same evaluation protocol.

*Half cat, Half woman, A princess, Highly detailed face, ultra-realistic, Jewelry.*

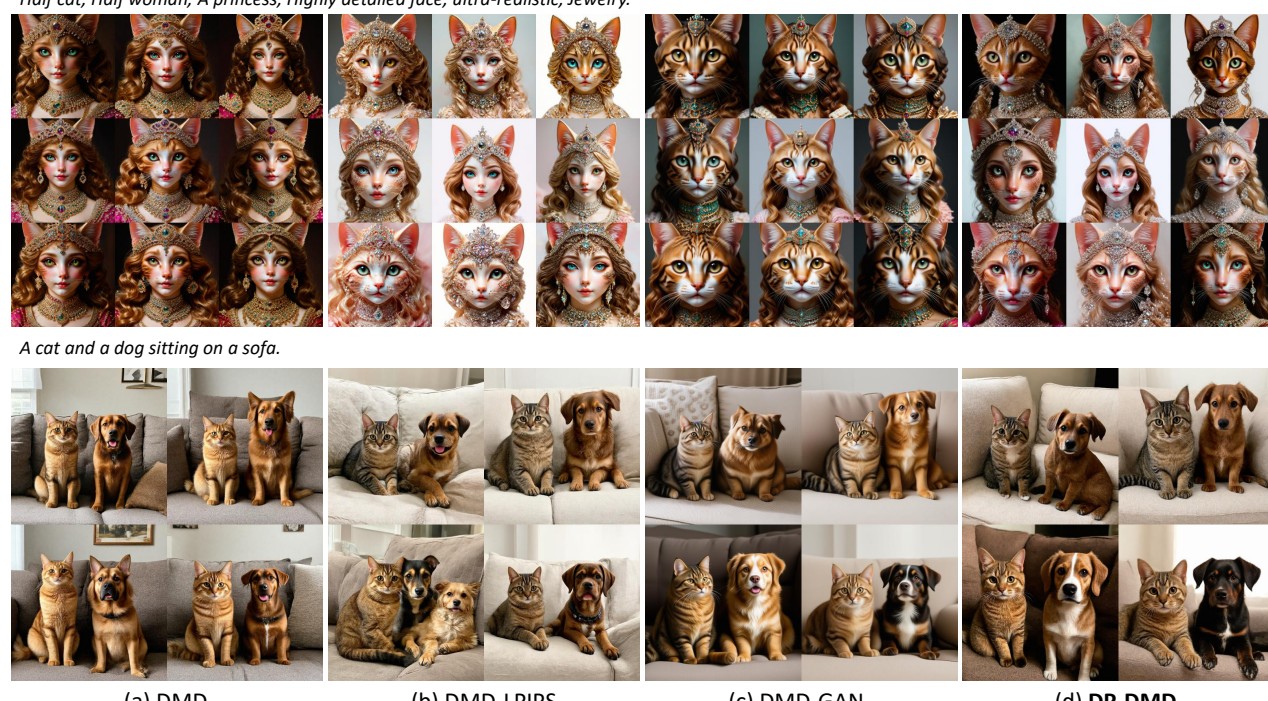

*A cat and a dog sitting on a sofa.*

| (a) DMD | (b) DMD-LPIPS | (c) DMD-GAN | (d) **DP-DMD** |

Figure 3. **Visual comparison of diversity supervision strategies**. Perceptual and adversarial regularization provide limited or less stable diversity gains and may introduce visual artifacts. In contrast, DP-DMD preserves richer prompt-conditioned variation while maintaining high perceptual quality, yielding a more favorable diversity-quality trade-off.

Table 3. **Effect of the diversity anchor step $K$.**

| $K$ | Diversity | | Quality | | Preference | |
|---|---|---|---|---|---|---|
| | DINO↑ | CLIP↑ | VQ-R1↑ | MIQA↑ | ImgR↑ | PicS↑ |
| Base | 0.137 | 0.133 | **4.649** | 1.016 | **1.189** | 21.75 |
| 1 | 0.157 | 0.155 | 4.578 | 0.995 | 1.121 | 21.62 |
| 3 | 0.175 | 0.178 | 4.648 | 1.016 | 1.125 | 21.67 |
| 5 | 0.179 | 0.182 | 4.646 | **1.017** | 1.142 | 21.76 |
| 10 | **0.187** | **0.185** | 4.589 | 1.004 | 1.112 | **21.78** |
| 30 | **0.187** | 0.181 | 4.602 | 1.007 | 1.117 | 21.71 |

Table 4. **Effect of the diversity weight $\lambda$.**

| $\lambda$ | Diversity | | Quality | | Preference | |
|---|---|---|---|---|---|---|
| | DINO↑ | CLIP↑ | VQ-R1↑ | MIQA↑ | ImgR↑ | PicS↑ |
| Base | 0.137 | 0.133 | 4.649 | **1.016** | **1.189** | 21.75 |
| 0.01 | 0.170 | 0.164 | **4.672** | 1.003 | 1.136 | **21.81** |
| 0.05 | 0.175 | **0.178** | 4.648 | **1.016** | 1.125 | 21.67 |
| 0.08 | 0.176 | 0.176 | 4.662 | 1.014 | 1.117 | 21.71 |
| 0.10 | **0.177** | 0.177 | 4.662 | 1.001 | 1.056 | 21.64 |

**Diversity anchor step $K$.** Table 3 studies where the teacher trajectory should be used to supervise the student's first distillation step. Adding the teacher-derived target at any tested anchor improves diversity over the DMD baseline, which supports the central premise that direct early-step supervision counteracts the support-narrowing behavior of DMD. Nevertheless, anchors that are too early are close to the noise prior and thus carry only weak semantic information; anchors that are too late mix diversity-relevant structure with teacher-specific refinements that the remaining few student steps may not reproduce reliably. Moderately early anchors offer the most useful compromise.

**Diversity weight $\lambda$.** Table 4 varies the relative strength of the first-step diversity loss. Increasing $\lambda$ strengthens the supervised connection between the initial noise and the teacher's early trajectory, thereby resisting the tendency of the DMD loss to concentrate probability mass around high-density, easily reproducible samples. Meanwhile, a large diversity weight can reduce the effective capacity allocated to final-image refinement. Moderate values achieve this separation most effectively.

**Gradient stopping.** Figure 5 verifies that role separation requires more than only two losses. Without stopping gradients after the first step, the DMD loss back-propagates through the entire student rollout, modifying the very mapping from noise to global structure of the image that the diversity loss is trying to preserve. The training curves show that this conflict appears early: the model quickly improves preference while losing diversity when the DMD

*Dark-haired Valerian and redhead Laureline, time and space agents, detailed and realistic painting by Carl Bloch.*

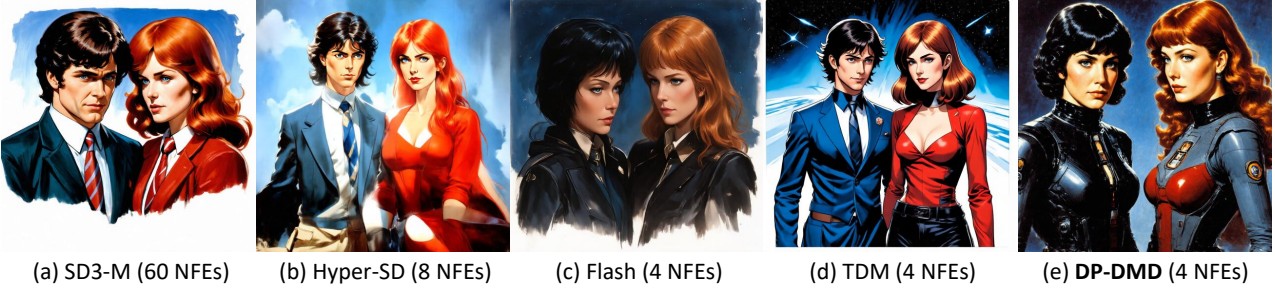

| (a) SD3-M (60 NFEs) | (b) Hyper-SD (8 NFEs) | (c) Flash (4 NFEs) | (d) TDM (4 NFEs) | (e) **DP-DMD** (4 NFEs) |

Figure 4. **Visual comparison** with open-source few-step distillation methods.

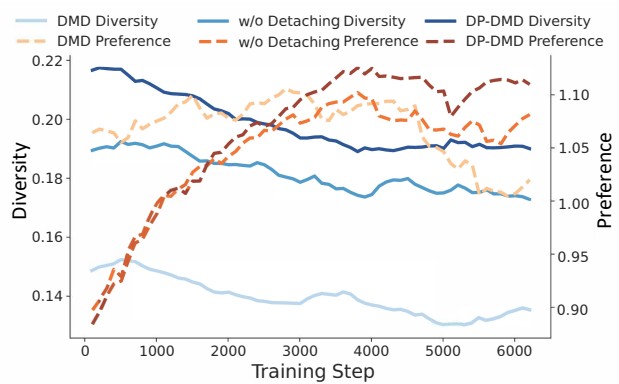

Figure 5. **Effect of gradient stopping**. Curves are plotted from the 100-th iteration and smoothed by moving average.

Table 5. **Prompt-following evaluation** on GenEval (Ghosh et al., 2023). We compare DP-DMD with the SD3.5-M teacher in terms of category-wise accuracies, with higher values indicating better prompt following.

| Method | Overall | Single Obj. | Two Obj. | Count. | Colors | Pos. | Color Attr. |
|---|---|---|---|---|---|---|---|
| SD3.5-M | **0.66** | 0.98 | 0.78 | **0.64** | **0.83** | 0.18 | **0.53** |
| **DP-DMD** | 0.65 | **0.99** | **0.81** | 0.60 | 0.81 | **0.21** | 0.48 |

gradient reaches the first step. Detaching resolves this loss interference: the two parts of the student are optimized for complementary functions rather than competing to control the same parameters through incompatible gradients.

**Prompt-following ability.** We additionally evaluate compositional prompt following on GenEval (Ghosh et al., 2023). This is to ensure that diversity preservation does not merely introduce random visual variation, but maintains the semantic alignment inherited from the teacher. As shown in Table 5, DP-DMD remains close to the teacher (SD3.5-M, Esser et al. 2024) overall and is particularly stable on object-presence and spatial-relation categories.

## 6. Conclusion and Discussion

We have presented DP-DMD, a simple role-separated distillation method for fast visual synthesis. DP-DMD separates diversity preservation and quality refinement across different denoising steps within the distilled student. The first step is supervised by a teacher-derived target-prediction loss to preserve diverse global modes, while the remaining steps are optimized with the standard DMD loss for perceptual quality refinement. Gradient stopping after the first step further prevents the mode-seeking DMD signal from overriding the diversity-preserving mapping.

DP-DMD currently applies explicit diversity supervision only to the first distillation step and uses a fixed anchor step and loss weight. While this simple design is effective, it may not be optimal when diversity-relevant decisions are distributed across multiple denoising steps, or when difficult prompts, strong guidance, or model-specific dynamics cause later stages to influence global structure. A promising direction is therefore to develop *adaptive role separation*, where the anchor location, diversity weight, and gradient routing are selected dynamically according to timestep, prompt complexity, teacher uncertainty, or student capacity. Such a design could extend the current first-step supervision into a trajectory-aware objective that preserves global variation while still allowing later steps to specialize in visual quality.

Beyond the reported evaluations, our preliminary studies suggest an additional practical advantage: the role-separated training yields more stable optimization when distilling from checkpoints at different stages of the model development pipeline, including pre-trained, mid-trained, and post-trained models via reinforcement learning. This robustness is important for real deployment scenarios, where foundation models are frequently updated and distillation must remain reliable across heterogeneous checkpoint states. These preliminary findings suggest that DP-DMD may serve not only as a diversity-preserving objective design, but also as a stable training recipe for iterative model development.

## Acknowledgments

This work was supported in part by the Hong Kong ITC Innovation and Technology Fund (9440379 and 9440390) and the PolyU-OPPO Joint Innovative Research Center.

## Impact Statement

This work contributes to the development of efficient generative modeling by improving diversity preservation in few-step model distillation. By enabling faster image synthesis while better maintaining the coverage of the teacher model distribution, the proposed method may support more practical deployment of high-quality text-to-image generators in resource-constrained or latency-sensitive settings.

All datasets used in this study are publicly available and are used in accordance with their respective licenses. As with other advances in image generation, potential downstream risks may include misuse for creating misleading, biased, or harmful visual content. These risks are not unique to the proposed method, but improved sampling efficiency may make responsible deployment practices increasingly important. We therefore encourage future use of this technique together with appropriate safeguards, such as dataset auditing, content moderation, watermarking, provenance tracking, and application-specific risk assessment.

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

# Appendix

## A. Derivation of the DMD Gradient

For completeness, we derive the DMD gradient used in Equation (5) of the main paper. Let

$$p_{\boldsymbol{\theta}}(\boldsymbol{x}) \triangleq p_{\text{stu}}(\boldsymbol{x}), \qquad q(\boldsymbol{x}) \triangleq p_{\text{tea}}(\boldsymbol{x}).$$

The DMD loss is the reverse KL divergence:

$$\ell_{\text{DMD}}(\boldsymbol{\theta}) = D_{\text{KL}}\big(p_{\boldsymbol{\theta}}(\boldsymbol{x}) \,\|\, q(\boldsymbol{x})\big) = \int p_{\boldsymbol{\theta}}(\boldsymbol{x}) \log \frac{p_{\boldsymbol{\theta}}(\boldsymbol{x})}{q(\boldsymbol{x})} \, d\boldsymbol{x}. \tag{10}$$

We now differentiate Equation (10) with respect to $\boldsymbol{\theta}$. The interchange between differentiation and integration is valid under standard regularity assumptions, which allows the use of the Leibniz integral rule, or equivalently, the dominated convergence theorem, to write

$$\nabla_{\boldsymbol{\theta}} \ell_{\text{DMD}}(\boldsymbol{\theta}) = \int \nabla_{\boldsymbol{\theta}} \left[ p_{\boldsymbol{\theta}}(\boldsymbol{x}) \log \frac{p_{\boldsymbol{\theta}}(\boldsymbol{x})}{q(\boldsymbol{x})} \right] d\boldsymbol{x}. \tag{11}$$

Since the teacher density $q(\boldsymbol{x})$ does not depend on $\boldsymbol{\theta}$, the integrand can be differentiated as

$$\begin{aligned}
\nabla_{\boldsymbol{\theta}} \left[ p_{\boldsymbol{\theta}}(\boldsymbol{x}) \log \frac{p_{\boldsymbol{\theta}}(\boldsymbol{x})}{q(\boldsymbol{x})} \right] &= \nabla_{\boldsymbol{\theta}} p_{\boldsymbol{\theta}}(\boldsymbol{x}) \log \frac{p_{\boldsymbol{\theta}}(\boldsymbol{x})}{q(\boldsymbol{x})} + p_{\boldsymbol{\theta}}(\boldsymbol{x}) \nabla_{\boldsymbol{\theta}} \log \frac{p_{\boldsymbol{\theta}}(\boldsymbol{x})}{q(\boldsymbol{x})} \\
&= \nabla_{\boldsymbol{\theta}} p_{\boldsymbol{\theta}}(\boldsymbol{x}) \log \frac{p_{\boldsymbol{\theta}}(\boldsymbol{x})}{q(\boldsymbol{x})} + p_{\boldsymbol{\theta}}(\boldsymbol{x}) \frac{\nabla_{\boldsymbol{\theta}} p_{\boldsymbol{\theta}}(\boldsymbol{x})}{p_{\boldsymbol{\theta}}(\boldsymbol{x})} \\
&= \nabla_{\boldsymbol{\theta}} p_{\boldsymbol{\theta}}(\boldsymbol{x}) \left( \log p_{\boldsymbol{\theta}}(\boldsymbol{x}) - \log q(\boldsymbol{x}) + 1 \right).
\end{aligned} \tag{12}$$

Substituting Equation (12) into Equation (11) gives

$$\nabla_{\boldsymbol{\theta}} \ell_{\text{DMD}}(\boldsymbol{\theta}) = \int \nabla_{\boldsymbol{\theta}} p_{\boldsymbol{\theta}}(\boldsymbol{x}) \left( \log p_{\boldsymbol{\theta}}(\boldsymbol{x}) - \log q(\boldsymbol{x}) + 1 \right) d\boldsymbol{x}. \tag{13}$$

Following the **continuity-equation** view of flow-based generative modeling (Lipman et al., 2022), we derive the identity that connects $\nabla_{\boldsymbol{\theta}} p_{\boldsymbol{\theta}}(\boldsymbol{x})$ to the velocity field induced in sample space by an infinitesimal parameter perturbation. Let the student generator be

$$\boldsymbol{g}_{\boldsymbol{\theta}} : \mathbb{R}^{D \times 1} \to \mathbb{R}^{M \times 1}, \qquad \boldsymbol{\epsilon} \in \mathbb{R}^{D \times 1}, \qquad \boldsymbol{x}_{\boldsymbol{\theta}} = \boldsymbol{g}_{\boldsymbol{\theta}}(\boldsymbol{\epsilon}) \in \mathbb{R}^{M \times 1},$$

where $\boldsymbol{\theta} \in \mathbb{R}^{L \times 1}$ denotes the student parameters and $\boldsymbol{\epsilon} \sim p_{\text{noise}}$ is sampled from a fixed noise distribution. Thus $\boldsymbol{x}_{\boldsymbol{\theta}} \sim p_{\boldsymbol{\theta}}$. Consider an infinitesimal parameter perturbation along an arbitrary direction $\boldsymbol{v} \in \mathbb{R}^{L \times 1}$:

$$\boldsymbol{\theta}_{\eta} = \boldsymbol{\theta} + \eta \boldsymbol{v}, \qquad \boldsymbol{x}_{\eta} = \boldsymbol{g}_{\boldsymbol{\theta}_{\eta}}(\boldsymbol{\epsilon}).$$

The Jacobian of the generator with respect to the parameters is

$$\nabla_{\boldsymbol{\theta}} \boldsymbol{g}_{\boldsymbol{\theta}}(\boldsymbol{\epsilon}) = \frac{\partial \boldsymbol{g}_{\boldsymbol{\theta}}(\boldsymbol{\epsilon})}{\partial \boldsymbol{\theta}} \in \mathbb{R}^{M \times L}.$$

Therefore, the infinitesimal motion of the generated sample in $\boldsymbol{x}$-space is

$$\frac{d\boldsymbol{x}_{\eta}}{d\eta} \bigg|_{\eta=0} = \nabla_{\boldsymbol{\theta}} \boldsymbol{g}_{\boldsymbol{\theta}+\eta\boldsymbol{v}}(\boldsymbol{\epsilon}) \boldsymbol{v} \big|_{\eta=0} = \nabla_{\boldsymbol{\theta}} \boldsymbol{g}_{\boldsymbol{\theta}}(\boldsymbol{\epsilon}) \boldsymbol{v} \in \mathbb{R}^{M \times 1}. \tag{14}$$

This perturbation induces a velocity field in sample space,

$$\boldsymbol{u}_{\boldsymbol{v}}(\boldsymbol{x}) = \mathbb{E}_{\boldsymbol{\epsilon} \sim p_{\text{noise}}(\cdot | \boldsymbol{x})} \left[ \nabla_{\boldsymbol{\theta}} \boldsymbol{g}_{\boldsymbol{\theta}}(\boldsymbol{\epsilon}) \boldsymbol{v} \mid \boldsymbol{g}_{\boldsymbol{\theta}}(\boldsymbol{\epsilon}) = \boldsymbol{x} \right] \in \mathbb{R}^{M \times 1},$$

where $p_{\text{noise}}(\cdot \mid \boldsymbol{x})$ denotes the conditional distribution of noise variables that generate the sample $\boldsymbol{x}$. The expectation is needed because $\boldsymbol{g}_{\boldsymbol{\theta}}$ need not be injective: multiple noise realizations may map to the same sample. Averaging over this conditional distribution therefore defines the mean sample-space velocity at $\boldsymbol{x}$ under the parameter perturbation $\boldsymbol{v}$. We

now derive how the density changes under this velocity field. Let $\varphi : \mathbb{R}^{M \times 1} \to \mathbb{R}$ be a smooth test function with compact support. Then $\nabla_{\boldsymbol{x}} \varphi(\boldsymbol{x}) \in \mathbb{R}^{M \times 1}$, and

$$
\begin{aligned}
\frac{d}{d\eta} \mathbb{E}_{\boldsymbol{x} \sim p_{\boldsymbol{\theta}_\eta}} [\varphi(\boldsymbol{x})] \bigg|_{\eta=0} &= \frac{d}{d\eta} \mathbb{E}_{\boldsymbol{\epsilon} \sim p_{\text{noise}}} [\varphi(\boldsymbol{g}_{\boldsymbol{\theta}_\eta}(\boldsymbol{\epsilon}))] \bigg|_{\eta=0} \\
&= \mathbb{E}_{\boldsymbol{\epsilon} \sim p_{\text{noise}}} \left[ \nabla_{\boldsymbol{x}} \varphi(\boldsymbol{g}_{\boldsymbol{\theta}}(\boldsymbol{\epsilon}))^{\mathsf{T}} \nabla_{\boldsymbol{\theta}} \boldsymbol{g}_{\boldsymbol{\theta}}(\boldsymbol{\epsilon}) \boldsymbol{v} \right] \\
&= \mathbb{E}_{\boldsymbol{x} \sim p_{\boldsymbol{\theta}}} \left[ \mathbb{E}_{\boldsymbol{\epsilon} \sim p_{\text{noise}}(\cdot | \boldsymbol{x})} \left[ \nabla_{\boldsymbol{x}} \varphi(\boldsymbol{x})^{\mathsf{T}} \nabla_{\boldsymbol{\theta}} \boldsymbol{g}_{\boldsymbol{\theta}}(\boldsymbol{\epsilon}) \boldsymbol{v} \right] \right] \\
&= \mathbb{E}_{\boldsymbol{x} \sim p_{\boldsymbol{\theta}}} \left[ \nabla_{\boldsymbol{x}} \varphi(\boldsymbol{x})^{\mathsf{T}} \mathbb{E}_{\boldsymbol{\epsilon} \sim p_{\text{noise}}(\cdot | \boldsymbol{x})} \left[ \nabla_{\boldsymbol{\theta}} \boldsymbol{g}_{\boldsymbol{\theta}}(\boldsymbol{\epsilon}) \boldsymbol{v} \right] \right] \\
&= \mathbb{E}_{\boldsymbol{x} \sim p_{\boldsymbol{\theta}}} \left[ \nabla_{\boldsymbol{x}} \varphi(\boldsymbol{x})^{\mathsf{T}} \boldsymbol{u}_{\boldsymbol{v}}(\boldsymbol{x}) \right] \\
&= \int p_{\boldsymbol{\theta}}(\boldsymbol{x}) \nabla_{\boldsymbol{x}} \varphi(\boldsymbol{x})^{\mathsf{T}} \boldsymbol{u}_{\boldsymbol{v}}(\boldsymbol{x}) \, d\boldsymbol{x}.
\end{aligned}
\tag{15}
$$

The third equality follows from the law of total expectation and the fourth equality holds because $\nabla_{\boldsymbol{x}} \varphi(\boldsymbol{x})$ is fixed once we condition on the generated sample location $\boldsymbol{x}$. Also note that the dimensions in the integrand are $p_{\boldsymbol{\theta}}(\boldsymbol{x}) \in \mathbb{R}$, $\nabla_{\boldsymbol{x}} \varphi(\boldsymbol{x})^{\mathsf{T}} \in \mathbb{R}^{1 \times M}$, and $\boldsymbol{u}_{\boldsymbol{v}}(\boldsymbol{x}) \in \mathbb{R}^{M \times 1}$, so the integrand is a scalar.

We continue to rewrite the right-hand side of Equation (15). Define the probability flux

$$
\boldsymbol{F}(\boldsymbol{x}) = p_{\boldsymbol{\theta}}(\boldsymbol{x}) \boldsymbol{u}_{\boldsymbol{v}}(\boldsymbol{x}).
$$

Since $\boldsymbol{x} \in \mathbb{R}^{M \times 1}$ and $\boldsymbol{u}_{\boldsymbol{v}}(\boldsymbol{x}) \in \mathbb{R}^{M \times 1}$, we have $\boldsymbol{F} : \mathbb{R}^{M \times 1} \to \mathbb{R}^{M \times 1}$. Thus the divergence of $\boldsymbol{F}$ is well defined and is given by

$$
\nabla_{\boldsymbol{x}} \cdot \boldsymbol{F}(\boldsymbol{x}) = \sum_{i=1}^{M} \frac{\partial F_i(\boldsymbol{x})}{\partial x_i}.
$$

Using **the product rule for divergence**, we have

$$
\begin{aligned}
\nabla_{\boldsymbol{x}} \cdot \left( \varphi(\boldsymbol{x}) \boldsymbol{F}(\boldsymbol{x}) \right) &= \sum_{i=1}^{M} \frac{\partial}{\partial x_i} \left( \varphi(\boldsymbol{x}) F_i(\boldsymbol{x}) \right) \\
&= \sum_{i=1}^{M} \left[ \frac{\partial \varphi(\boldsymbol{x})}{\partial x_i} F_i(\boldsymbol{x}) + \varphi(\boldsymbol{x}) \frac{\partial F_i(\boldsymbol{x})}{\partial x_i} \right] \\
&= \nabla_{\boldsymbol{x}} \varphi(\boldsymbol{x})^{\mathsf{T}} \boldsymbol{F}(\boldsymbol{x}) + \varphi(\boldsymbol{x}) \nabla_{\boldsymbol{x}} \cdot \boldsymbol{F}(\boldsymbol{x}).
\end{aligned}
\tag{16}
$$

Rearranging Equation (16) gives

$$
\nabla_{\boldsymbol{x}} \varphi(\boldsymbol{x})^{\mathsf{T}} \boldsymbol{F}(\boldsymbol{x}) = \nabla_{\boldsymbol{x}} \cdot \left( \varphi(\boldsymbol{x}) \boldsymbol{F}(\boldsymbol{x}) \right) - \varphi(\boldsymbol{x}) \nabla_{\boldsymbol{x}} \cdot \boldsymbol{F}(\boldsymbol{x}).
\tag{17}
$$

Integrating Equation (17) over a bounded domain $\Omega \subset \mathbb{R}^M$ yields

$$
\int_{\Omega} \nabla_{\boldsymbol{x}} \varphi(\boldsymbol{x})^{\mathsf{T}} \boldsymbol{F}(\boldsymbol{x}) \, d\boldsymbol{x} = \int_{\Omega} \nabla_{\boldsymbol{x}} \cdot \left( \varphi(\boldsymbol{x}) \boldsymbol{F}(\boldsymbol{x}) \right) d\boldsymbol{x} - \int_{\Omega} \varphi(\boldsymbol{x}) \nabla_{\boldsymbol{x}} \cdot \boldsymbol{F}(\boldsymbol{x}) \, d\boldsymbol{x}.
\tag{18}
$$

By the divergence theorem,

$$
\int_{\Omega} \nabla_{\boldsymbol{x}} \cdot \left( \varphi(\boldsymbol{x}) \boldsymbol{F}(\boldsymbol{x}) \right) d\boldsymbol{x} = \int_{\partial\Omega} \varphi(\boldsymbol{x}) \boldsymbol{F}(\boldsymbol{x})^{\mathsf{T}} \boldsymbol{n}(\boldsymbol{x}) \, dS,
$$

where $\boldsymbol{n}(\boldsymbol{x})$ is the outward unit normal on the boundary $\partial\Omega$. The boundary term vanishes and thus can be dropped under the standard assumption that $\varphi$ has compact support. Hence,

$$
\int \nabla_{\boldsymbol{x}} \varphi(\boldsymbol{x})^{\mathsf{T}} \boldsymbol{F}(\boldsymbol{x}) \, d\boldsymbol{x} = - \int \varphi(\boldsymbol{x}) \nabla_{\boldsymbol{x}} \cdot \boldsymbol{F}(\boldsymbol{x}) \, d\boldsymbol{x}.
\tag{19}
$$

Substituting $\boldsymbol{F}(\boldsymbol{x}) = p_{\boldsymbol{\theta}}(\boldsymbol{x}) \boldsymbol{u}_{\boldsymbol{v}}(\boldsymbol{x})$ into Equation (19), we obtain

$$
\int \nabla_{\boldsymbol{x}} \varphi(\boldsymbol{x})^{\mathsf{T}} \left( p_{\boldsymbol{\theta}}(\boldsymbol{x}) \boldsymbol{u}_{\boldsymbol{v}}(\boldsymbol{x}) \right) d\boldsymbol{x} = - \int \varphi(\boldsymbol{x}) \nabla_{\boldsymbol{x}} \cdot \left( p_{\boldsymbol{\theta}}(\boldsymbol{x}) \boldsymbol{u}_{\boldsymbol{v}}(\boldsymbol{x}) \right) d\boldsymbol{x}.
\tag{20}
$$

On the other hand, differentiating the same expectation in Equation (15) through the density gives

$$\frac{d}{d\eta}\mathbb{E}_{\boldsymbol{x}\sim p_{\boldsymbol{\theta}_\eta}}[\varphi(\boldsymbol{x})]\bigg|_{\eta=0} = \frac{d}{d\eta}\int \varphi(\boldsymbol{x})p_{\boldsymbol{\theta}+\eta\boldsymbol{v}}(\boldsymbol{x})\,d\boldsymbol{x}\bigg|_{\eta=0} = \int \varphi(\boldsymbol{x})\frac{d}{d\eta}p_{\boldsymbol{\theta}+\eta\boldsymbol{v}}(\boldsymbol{x})\bigg|_{\eta=0}\,d\boldsymbol{x}. \tag{21}$$

Comparing Equation (21) with Equation (20) and noting that the equality holds for every smooth compactly supported test function $\varphi$, we obtain the weak-form identity

$$\boxed{\frac{d}{d\eta}p_{\boldsymbol{\theta}+\eta\boldsymbol{v}}(\boldsymbol{x})\bigg|_{\eta=0} = -\nabla_{\boldsymbol{x}}\cdot(p_{\boldsymbol{\theta}}(\boldsymbol{x})\boldsymbol{u}_{\boldsymbol{v}}(\boldsymbol{x}))} \tag{22}$$

Equation (22) is the continuity equation induced by the parameter perturbation direction $\boldsymbol{v}$. Finally, because $\boldsymbol{v}\in\mathbb{R}^{L\times 1}$ is arbitrary, the full parameter gradient can be recovered component-wise. For the $j$-th parameter coordinate, choose $\boldsymbol{v}=\boldsymbol{e}_j$, where $\boldsymbol{e}_j$ is the $j$-th standard basis vector in $\mathbb{R}^{L\times 1}$. Then

$$\frac{\partial p_{\boldsymbol{\theta}}(\boldsymbol{x})}{\partial\theta_j} = -\nabla_{\boldsymbol{x}}\cdot\left(p_{\boldsymbol{\theta}}(\boldsymbol{x})\boldsymbol{u}_{\boldsymbol{e}_j}(\boldsymbol{x})\right), \tag{23}$$

where

$$\boldsymbol{u}_{\boldsymbol{e}_j}(\boldsymbol{x}) = \mathbb{E}_{\boldsymbol{\epsilon}\sim p_{\text{noise}}(\cdot|\boldsymbol{x})}\left[\nabla_{\boldsymbol{\theta}}\boldsymbol{g}_{\boldsymbol{\theta}}(\boldsymbol{\epsilon})\boldsymbol{e}_j \mid \boldsymbol{g}_{\boldsymbol{\theta}}(\boldsymbol{\epsilon}) = \boldsymbol{x}\right] = \mathbb{E}_{\boldsymbol{\epsilon}\sim p_{\text{noise}}(\cdot|\boldsymbol{x})}\left[\frac{\partial\boldsymbol{g}_{\boldsymbol{\theta}}(\boldsymbol{\epsilon})}{\partial\theta_j} \;\middle|\; \boldsymbol{g}_{\boldsymbol{\theta}}(\boldsymbol{\epsilon}) = \boldsymbol{x}\right] \in \mathbb{R}^{M\times 1}. \tag{24}$$

Thus, the compact notation

$$\boxed{\nabla_{\boldsymbol{\theta}}p_{\boldsymbol{\theta}}(\boldsymbol{x}) = -\nabla_{\boldsymbol{x}}\cdot(p_{\boldsymbol{\theta}}(\boldsymbol{x})\boldsymbol{U}_{\boldsymbol{\theta}}(\boldsymbol{x}))} \tag{25}$$

should be interpreted component-wise: the left-hand side is a vector in $\mathbb{R}^{L\times 1}$, and the right-hand side collects the $L$ divergence terms induced by the $L$ velocity fields, *i.e.*, $\boldsymbol{U}_{\boldsymbol{\theta}}(\boldsymbol{x}) = [\boldsymbol{u}_{\boldsymbol{e}_1}(\boldsymbol{x}),\ldots,\boldsymbol{u}_{\boldsymbol{e}_L}(\boldsymbol{x})]\in\mathbb{R}^{M\times L}$. Let

$$\varphi(\boldsymbol{x}) = \log p_{\boldsymbol{\theta}}(\boldsymbol{x}) - \log q(\boldsymbol{x}) + 1. \tag{26}$$

Substituting Equations (25) and (26) into Equation (13), we obtain

$$\nabla_{\boldsymbol{\theta}}\ell_{\text{DMD}}(\boldsymbol{\theta}) = -\int \varphi(\boldsymbol{x})\nabla_{\boldsymbol{x}}\cdot(p_{\boldsymbol{\theta}}(\boldsymbol{x})\boldsymbol{U}_{\boldsymbol{\theta}}(\boldsymbol{x}))\,d\boldsymbol{x}. \tag{27}$$

We now apply integration by parts component-wise. For the $j$-th parameter coordinate, define

$$\boldsymbol{V}_j(\boldsymbol{x}) = p_{\boldsymbol{\theta}}(\boldsymbol{x})\boldsymbol{u}_{\boldsymbol{e}_j}(\boldsymbol{x}) \in \mathbb{R}^{M\times 1}.$$

Then

$$\begin{aligned}
\frac{\partial\ell_{\text{DMD}}(\boldsymbol{\theta})}{\partial\theta_j} &= -\int_{\Omega} \varphi(\boldsymbol{x})\nabla_{\boldsymbol{x}}\cdot\boldsymbol{V}_j(\boldsymbol{x})\,d\boldsymbol{x} \\
&= -\int_{\Omega}\left[\nabla_{\boldsymbol{x}}\cdot(\varphi(\boldsymbol{x})\boldsymbol{V}_j(\boldsymbol{x})) - \nabla_{\boldsymbol{x}}\varphi(\boldsymbol{x})^{\mathsf{T}}\boldsymbol{V}_j(\boldsymbol{x})\right]d\boldsymbol{x} \\
&= -\int_{\partial\Omega} \varphi(\boldsymbol{x})\boldsymbol{V}_j(\boldsymbol{x})^{\mathsf{T}}\boldsymbol{n}(\boldsymbol{x})\,dS + \int_{\Omega}\nabla_{\boldsymbol{x}}\varphi(\boldsymbol{x})^{\mathsf{T}}\boldsymbol{V}_j(\boldsymbol{x})\,d\boldsymbol{x},
\end{aligned} \tag{28}$$

where the second equality follows from the product rule for divergence in Equation (16) and the third equality follows from the divergence theorem. We once again assume that the boundary term in Equation (28) vanishes. Thus, for the $j$-th parameter coordinate, we have

$$\frac{\partial\ell_{\text{DMD}}(\boldsymbol{\theta})}{\partial\theta_j} = \int p_{\boldsymbol{\theta}}(\boldsymbol{x})\boldsymbol{u}_{\boldsymbol{e}_j}(\boldsymbol{x})^{\mathsf{T}}\nabla_{\boldsymbol{x}}\varphi(\boldsymbol{x})\,d\boldsymbol{x}. \tag{29}$$

Substituting the definition of $\varphi(\boldsymbol{x})$ into Equation (29) gives

$$\begin{aligned}
\frac{\partial\ell_{\text{DMD}}(\boldsymbol{\theta})}{\partial\theta_j} &= \int p_{\boldsymbol{\theta}}(\boldsymbol{x})\boldsymbol{u}_{\boldsymbol{e}_j}(\boldsymbol{x})^{\mathsf{T}}\nabla_{\boldsymbol{x}}\left(\log p_{\boldsymbol{\theta}}(\boldsymbol{x}) - \log q(\boldsymbol{x}) + 1\right)d\boldsymbol{x} \\
&= \int p_{\boldsymbol{\theta}}(\boldsymbol{x})\boldsymbol{u}_{\boldsymbol{e}_j}(\boldsymbol{x})^{\mathsf{T}}\left(\nabla_{\boldsymbol{x}}\log p_{\boldsymbol{\theta}}(\boldsymbol{x}) - \nabla_{\boldsymbol{x}}\log q(\boldsymbol{x})\right)d\boldsymbol{x},
\end{aligned} \tag{30}$$

where the constant term disappears because $\nabla_{\boldsymbol{x}} 1 = \boldsymbol{0}$. Define the student and teacher scores as

$$\boldsymbol{s}_{\text{stu}}(\boldsymbol{x}) = \nabla_{\boldsymbol{x}} \log p_{\boldsymbol{\theta}}(\boldsymbol{x}), \qquad \boldsymbol{s}_{\text{tea}}(\boldsymbol{x}) = \nabla_{\boldsymbol{x}} \log q(\boldsymbol{x}), \tag{31}$$

and define their difference by

$$\Delta\boldsymbol{s}(\boldsymbol{x}) = \boldsymbol{s}_{\text{stu}}(\boldsymbol{x}) - \boldsymbol{s}_{\text{tea}}(\boldsymbol{x}). \tag{32}$$

Then Equation (30) can be written as

$$\frac{\partial \ell_{\text{DMD}}(\boldsymbol{\theta})}{\partial \theta_j} = \mathbb{E}_{\boldsymbol{x} \sim p_{\boldsymbol{\theta}}} \left[ \boldsymbol{u}_{\boldsymbol{e}_j}(\boldsymbol{x})^{\mathsf{T}} \Delta\boldsymbol{s}(\boldsymbol{x}) \right]. \tag{33}$$

We now express the coordinate-wise velocity through the reparameterized generator using the law of total expectation:

$$\mathbb{E}_{\boldsymbol{x} \sim p_{\boldsymbol{\theta}}} \left[ \boldsymbol{u}_{\boldsymbol{e}_j}(\boldsymbol{x})^{\mathsf{T}} \Delta\boldsymbol{s}(\boldsymbol{x}) \right]$$
$$= \mathbb{E}_{\boldsymbol{x} \sim p_{\boldsymbol{\theta}}} \left[ \mathbb{E}_{\boldsymbol{\epsilon} \sim p_{\text{noise}}(\cdot|\boldsymbol{x})} \left[ \frac{\partial \boldsymbol{g}_{\boldsymbol{\theta}}(\boldsymbol{\epsilon})}{\partial \theta_j} \,\middle|\, \boldsymbol{g}_{\boldsymbol{\theta}}(\boldsymbol{\epsilon}) = \boldsymbol{x} \right]^{\mathsf{T}} \Delta\boldsymbol{s}(\boldsymbol{x}) \right]$$
$$= \mathbb{E}_{\boldsymbol{x} \sim p_{\boldsymbol{\theta}}} \left[ \mathbb{E}_{\boldsymbol{\epsilon} \sim p_{\text{noise}}(\cdot|\boldsymbol{x})} \left[ \frac{\partial \boldsymbol{g}_{\boldsymbol{\theta}}(\boldsymbol{\epsilon})}{\partial \theta_j}^{\mathsf{T}} \Delta\boldsymbol{s}(\boldsymbol{g}_{\boldsymbol{\theta}}(\boldsymbol{\epsilon})) \,\middle|\, \boldsymbol{g}_{\boldsymbol{\theta}}(\boldsymbol{\epsilon}) = \boldsymbol{x} \right] \right]$$
$$= \mathbb{E}_{\boldsymbol{\epsilon} \sim p_{\text{noise}}} \left[ \frac{\partial \boldsymbol{g}_{\boldsymbol{\theta}}(\boldsymbol{\epsilon})}{\partial \theta_j}^{\mathsf{T}} \Delta\boldsymbol{s}(\boldsymbol{g}_{\boldsymbol{\theta}}(\boldsymbol{\epsilon})) \right]. \tag{34}$$

Equivalently, since $\boldsymbol{x}_{\boldsymbol{\theta}} = \boldsymbol{g}_{\boldsymbol{\theta}}(\boldsymbol{\epsilon})$,

$$\frac{\partial \ell_{\text{DMD}}(\boldsymbol{\theta})}{\partial \theta_j} = \mathbb{E}_{\boldsymbol{x}_{\boldsymbol{\theta}}} \left[ \frac{\partial \boldsymbol{x}_{\boldsymbol{\theta}}}{\partial \theta_j}^{\mathsf{T}} \Delta\boldsymbol{s}(\boldsymbol{x}_{\boldsymbol{\theta}}) \right]. \tag{35}$$

Stacking Equation (35) over $j = 1, \ldots, L$ gives the vector form

$$\nabla_{\boldsymbol{\theta}} \ell_{\text{DMD}}(\boldsymbol{\theta}) = \mathbb{E}_{\boldsymbol{x}_{\boldsymbol{\theta}}} \left[ (\nabla_{\boldsymbol{\theta}} \boldsymbol{x}_{\boldsymbol{\theta}})^{\mathsf{T}} \Delta\boldsymbol{s}(\boldsymbol{x}_{\boldsymbol{\theta}}) \right]. \tag{36}$$

In practice, DMD evaluates the score difference in a perturbed space: the generated sample $\boldsymbol{x}_{\boldsymbol{\theta}}$ is diffused to $\boldsymbol{z}_t$, and the student and teacher scores are evaluated at $\boldsymbol{z}_t$. With a slight abuse of notation, we write

$$\Delta\boldsymbol{s}(\boldsymbol{z}_t) = \boldsymbol{s}_{\text{stu}}(\boldsymbol{z}_t) - \boldsymbol{s}_{\text{tea}}(\boldsymbol{z}_t). \tag{37}$$

Equation (36) then becomes

$$\nabla_{\boldsymbol{\theta}} \ell_{\text{DMD}}(\boldsymbol{\theta}) = \mathbb{E}_{\boldsymbol{x}_{\boldsymbol{\theta}}} \left[ (\nabla_{\boldsymbol{\theta}} \boldsymbol{x}_{\boldsymbol{\theta}})^{\mathsf{T}} \left( \boldsymbol{s}_{\text{stu}}(\boldsymbol{z}_t) - \boldsymbol{s}_{\text{tea}}(\boldsymbol{z}_t) \right) \right], \tag{38}$$

which recovers Equation (5) in the main paper.

## B. Observation of Early and Late Denoising Steps

Figure A visualizes the inference trajectory of SD3.5-M (Esser et al., 2024) under different noise initializations. The results reveal clear stage-wise denoising behavior. At early timesteps, where the latent remains highly noisy, the model already determines the global layout of the image, including overall composition, coarse geometry, and object identity. Differences introduced at this stage are propagated through the remaining trajectories, and lead to distinct final samples under the same text condition. This indicates that early denoising steps are closely tied to sample diversity.

By contrast, later timesteps mainly refine local appearance, such as texture, colors, contours, and fine details. These refinements improve perceptual quality but have a comparatively weaker effect on the global structure of the image. The right panel of Figure A further supports this observation: different noise initializations produce visibly different coarse structure before fine details emerge. This motivates the role-separated design of DP-DMD. The first distillation step is assigned a diversity-preserving objective that follows the teacher's early trajectories, while the remaining steps are optimized by DMD for quality refinement.

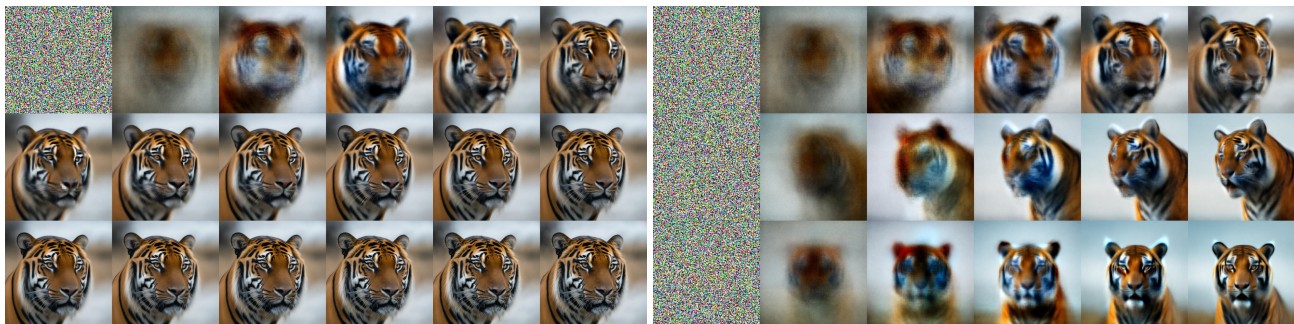

Figure A. **Progressive denoising dynamics**. SD3.5-M exhibits stage-wise generation behavior. The left panel shows one denoising trajectory from early to late timesteps. The right panel compares early denoising states from different noise initializations. Early steps establish diverse global structure, whereas later steps mainly refine local appearance.

## C. DP-DMD for Diffusion Models

The main paper formulates DP-DMD for flow-based models. For diffusion-based teachers such as SDXL (Podell et al., 2023), which are commonly parameterized by noise prediction, we define the diversity target in the denoised latent space, *i.e.*, the $\boldsymbol{x}$-prediction space. Starting from the same initial noisy latent $\boldsymbol{z}_T$, we run the teacher for $K$ denoising steps and obtain an intermediate latent $\boldsymbol{z}_{\tilde{t}}$. Given the teacher noise prediction $\boldsymbol{\epsilon}_{\text{tea}}(\boldsymbol{z}_{\tilde{t}}, \tilde{t})$, the teacher-implied denoised target is

$$\tilde{\boldsymbol{x}} = \frac{\boldsymbol{z}_{\tilde{t}} - \sqrt{1 - \alpha_{\tilde{t}}}\,\boldsymbol{\epsilon}_{\text{tea}}(\boldsymbol{z}_{\tilde{t}}, \tilde{t})}{\sqrt{\alpha_{\tilde{t}}}}, \tag{39}$$

where $\alpha_{\tilde{t}}$ denotes the cumulative noise-schedule coefficient. This target is the diffusion-model analogue of the teacher-derived intermediate state in Equation (6). For the student, the first-step denoised prediction from $\boldsymbol{z}_T$ is

$$\boldsymbol{x}_{\boldsymbol{\theta}}(\boldsymbol{z}_T, 0) = \frac{\boldsymbol{z}_T - \sqrt{1 - \alpha_T}\,\boldsymbol{\epsilon}_{\boldsymbol{\theta}}(\boldsymbol{z}_T, T)}{\sqrt{\alpha_T}}. \tag{40}$$

The diffusion-model diversity loss is then

$$\ell_{\text{Div}}(\boldsymbol{\theta}) = \mathbb{E}_{\boldsymbol{z}_T}\left[\|\boldsymbol{x}_{\boldsymbol{\theta}}(\boldsymbol{z}_T, 0) - \tilde{\boldsymbol{x}}\|^2\right]. \tag{41}$$

After this, the first-step output is detached, and the remaining $N - 1$ student steps are optimized with standard DMD.

## D. Subjective User Study

We conduct a controlled subjective user study to complement the quantitative results. We randomly sample 50 prompts and recruit 10 participants with experience evaluating image-generation results. For each prompt, outputs from two methods are shown side by side in randomized spatial order under the same text condition and random-seed protocol.

Participants compare the two methods along two criteria: sample diversity and image quality. Diversity emphasizes variation in overall composition, global and local structure, and semantic attributes across multiple samples from the same prompt. Image quality focuses on semantic and statistical fidelity and naturalness. Results are aggregated as pairwise win rates over all prompts and participants.

As shown in Figure B, DP-DMD is preferred over DMD (Yin et al., 2024a), DMD-LPIPS, and DMD-GAN in sample diversity, while maintaining competitive or superior image quality. These results support the central claim that role-separated distillation mitigates diversity collapse at no perceptual cost.

## E. Additional Qualitative Results

Figure C compares samples generated from identical prompts with different random seeds. DP-DMD produces broader variation in global layout, object appearance, and semantic attributes than other DMD-based baselines, confirming that its diversity gains correspond to meaningful sample variation rather than visual artifacts.

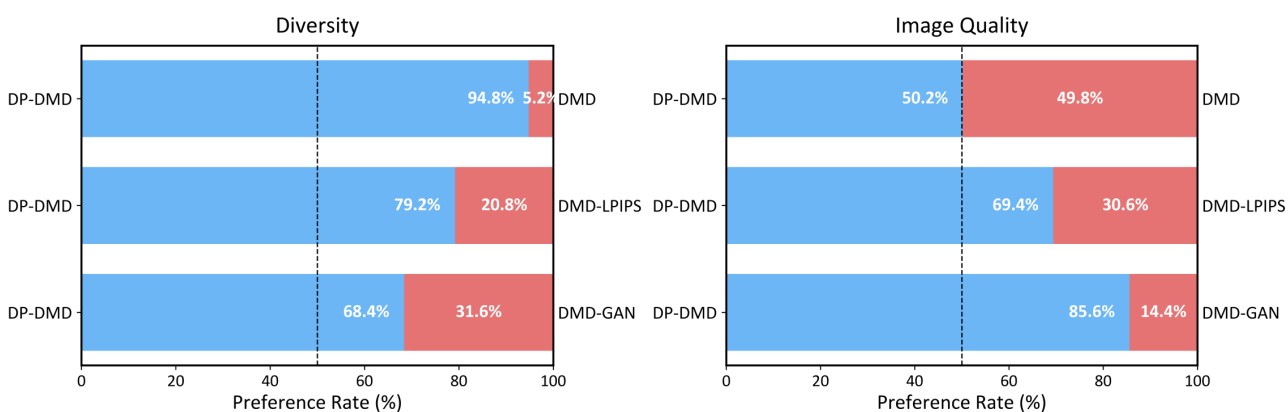

Figure B. **Subjective user study on sample diversity and image quality**. We report pairwise win rates of DP-DMD against DMD (Yin et al., 2024a), DMD-LPIPS, and DMD-GAN, evaluated on 50 prompts and by 10 participants. The dashed line marks the 50% chance-level win rate. DP-DMD is consistently preferred for sample diversity while remaining competitive image quality.

Figure D shows samples generated by DP-DMD at $1,024 \times 1,024$ resolution using 4 NFEs. The results demonstrate that the proposed diversity-preserving supervision does not prevent the student from producing coherent layouts, realistic appearance, and fine-grained details under few-step inference.

*A man with sunglasses.*

*A living room has a couch, a chair, and a fireplace.*

*A very cute dog wearing some type of silly hat.*

*A picture of a cute anime girl.*

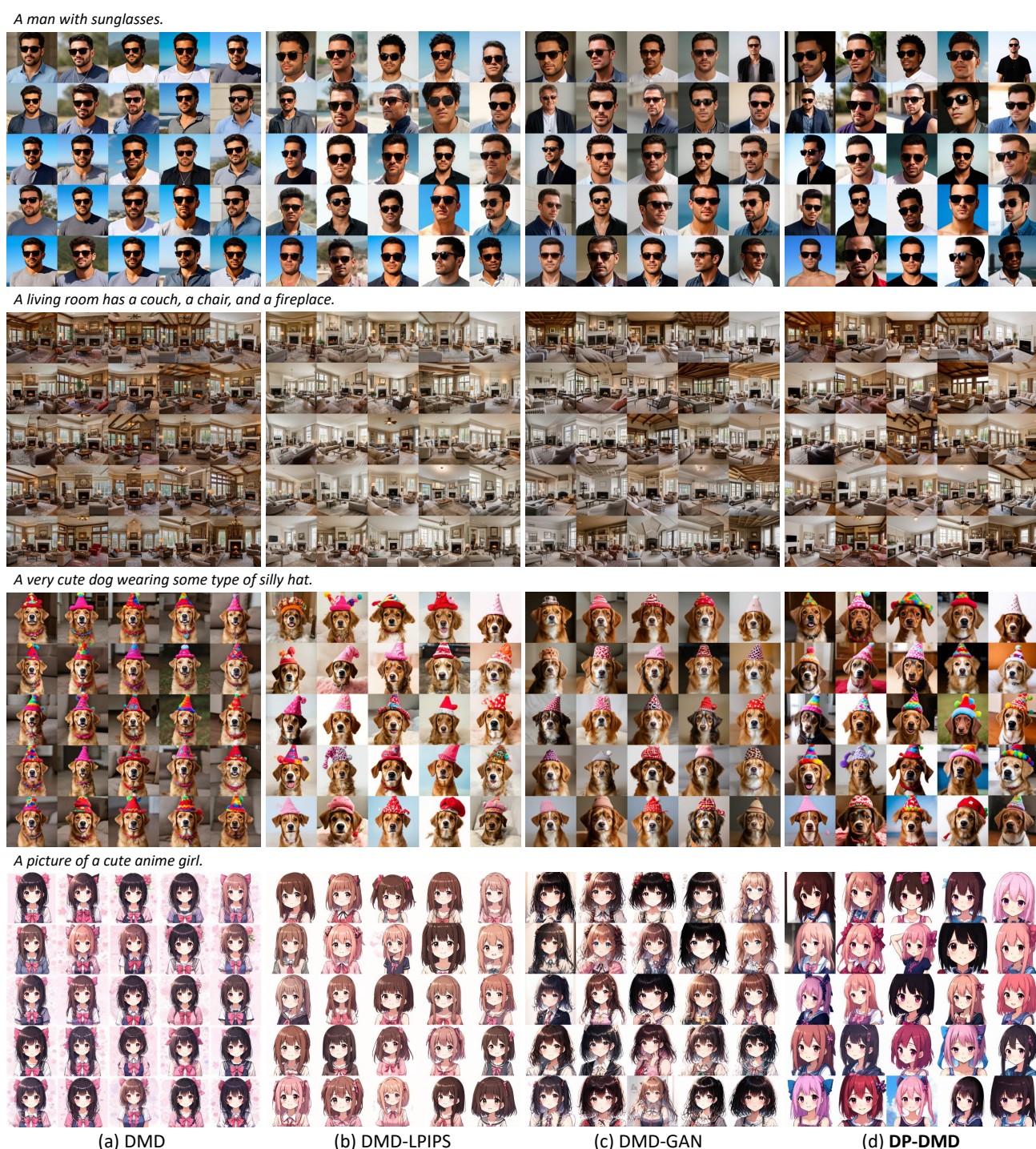

(a) DMD      (b) DMD-LPIPS      (c) DMD-GAN      (d) **DP-DMD**

Figure C. **Sample diversity under identical prompts**. Images are generated from the same text prompts and using different random seeds. DP-DMD produces richer global and semantic variation than the baselines while requiring only 4 NFEs.

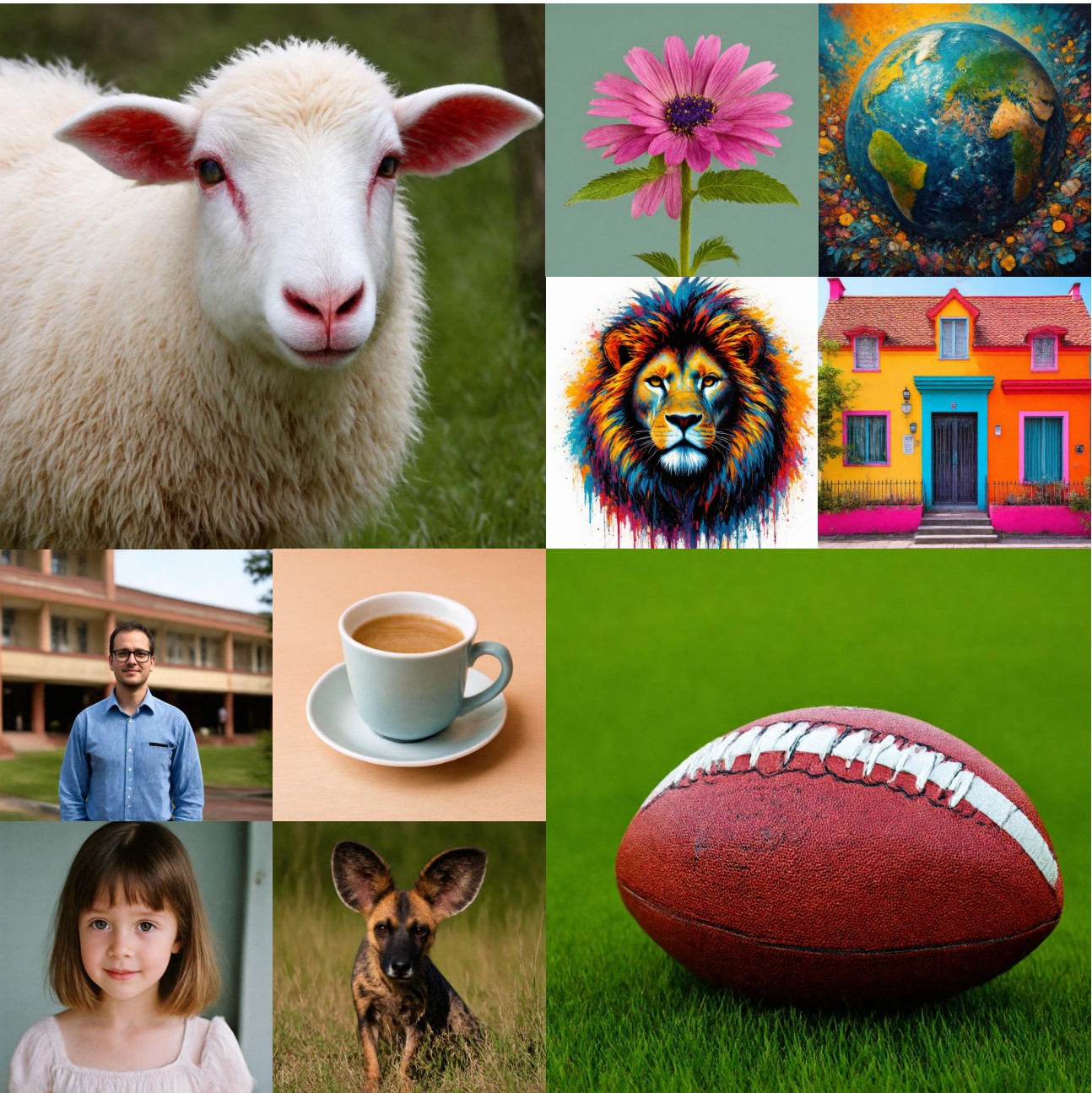

Figure D. **Sample quality of DP-DMD**. Images are generated at $1,024 \times 1,024$ resolution using DP-DMD distilled from SD3.5-M (Esser et al., 2024). All samples are produced with 4 NFEs.

