# OpenReview forum: "Diversity-Preserved Distribution Matching Distillation for Fast Visual Synthesis"
_ICML.cc/2026/Conference — ICML 2026 regular_

### Official Review · Reviewer_xLzh · 2026-03-08

**Soundness:** 3
**Presentation:** 3
**Significance:** 3
**Originality:** 3
**Overall Recommendation:** 5
**Confidence:** 3

**Summary:**

This paper proposes a new distillation method to improve diversity in the generated images. It builds on top of the established method distributiona matching distillation (DMD) by separating thes role of different diffusion steps. The first step of the distilled model is dedicated for diversity preservation by a supervised velocity target, and the following steps are for refining visual details using the original DMD loss. The stop gradient between the diffusion steps help to disentangle the effectiveness of the proposed loss and thus enhance diversity. The experimental results show the effectiveness of the proposed method in increasing the diversity, while preserving the visual quality and prompt following capability of the pre-trained model.

**Compliance With Llm Reviewing Policy:**

Affirmed.

**Key Questions For Authors:**

1. In the ablation study section, the best performed K or lambda are different given different properties or given different metrics used for a certain property. Can the authors explain how they choose K and lambda for different benchmarks, CFG scale, etc.?

2. In Algorithm 1, the fake student model update is missing.

3. In Figure 3, why the preference decreased when training longer with DMD? It seems that training longer will hurt preference while the diversity remains stable, does this pattern hold across benchmarks and preference metrics? How do the authors decide when to stop training?

**Limitations:**

yes

**Strengths And Weaknesses:**

Strengths:

1. The method is well motivated and addresses an important problem (mode collapse) in distillation.
2. The method is easy to implement, without the need of perceptual backbone, discriminator, auxiliary networks or additional ground-truth images.
3. The experimental section is thorough, covering a wide range of baselines, metrics, and benchmarks.

Weaknesses:

1. The presentation of the background is not friendly to readers who are not familiar with DMD method. It is not clear what is the fake model and how the fake model is trained.
2. Although the proposed method does not need perceptual backbone, discriminator, auxiliary networks or additional ground-truth images, choosing a good set of hyperparameters seem non-trivial. It is not clear how the hyperparams are chosen across datasets / benchmarks. See questions below.

---

> ### Author Rebuttal · Authors · 2026-03-27
>
> We are deeply grateful for the reviewer’s thoughtful insights and thorough evaluation of our manuscript. Please find what below our detailed point-to-point responses to all the comments of this reviewer. We hope our responses can address this reviewer's concerns.
>
> ---
>
> **`[W1 Regarding The Presentation]`**
>
> Thanks for the valuable comment. In the revised version, we will provide a clearer and more self-contained introduction to DMD, including a precise explanation of the “fake model” and its training procedure.
>
> ---
>
> **`[W2 & Q1 Regarding Hyperparameter Selection]`**
>
> Thanks for raising this important point. Our hyperparameter selection is as follows.
>
> - **On the choice of K:** The diversity anchor step K determines how far the teacher is rolled out to obtain the intermediate state. A larger K corresponds to a more denoised (semantically clearer) state, which provides stronger structural guidance for diversity preservation. However, increasing K also introduces additional computational overhead due to longer teacher rollouts, which becomes increasingly unfavorable for larger-scale models. Empirically, our ablation (Table 1) shows that even K=1 already provides a significant diversity improvement over the DMD baseline. Increasing K to 3 or 5 further improves diversity with diminishing returns. Based on this observation, we recommend using **small K values (e.g., K=3) in practice**, which strike a good balance between **diversity improvement and training efficiency**, especially for large models.
>
> - **On the choice of $\lambda$:** The weighting coefficient $\lambda$ controls the balance between diversity preservation and quality refinement. A smaller $\lambda$ allows the DMD objective (which favors high-quality samples) to dominate, while a larger $\lambda$ enforces stronger diversity regularization. This behavior is consistent with the trade-off observed in Table 2. In practice, we find that **$\lambda$ in the range of 0.01-0.05 works reliably across settings**. If higher visual quality is desired, a smaller value can be used; if stronger diversity is preferred, a slightly larger value is beneficial. Notably, this range is relatively stable and does not require dataset-specific tuning.
>
> - **On CFG and cross-benchmark settings:** Please kindly refer to our response to **`Reviewer V512 [W2]`**. It is observed that increasing CFG improves generation quality under DP-DMD **without degrading diversity**, indicating that our method is compatible with different guidance strengths. Importantly, we do **not perform benchmark-specific hyperparameter tuning**, as our goal is not to maximize performance through extensive tuning, but to provide a simple and practical solution to mitigate diversity collapse.
>
> We will include the details of hyperparameter selection in the revised manuscript.
>
> ---
>
> **`[Q2 Regarding Algorithm 1]`**
>
> Thanks for pointing out this issue. The current Algorithm 1 focuses on optimizing the student parameters, and we will add the update of fake model in the revised version.
>
> ---
>
> **`[Q3 Regarding Longer Training in DMD]`**
>
> We thank the reviewer for the insightful question. The decrease in preference under prolonged DMD training is mainly due to the **mode-seeking nature of the reverse-KL objective**, which gradually concentrates probability mass and leads to partial mode collapse. Empirically, early-stage models produce more diverse samples but may contain subtle local artifacts (e.g., checkerboard patterns), while later-stage models remove these artifacts but converge to simpler global patterns (e.g., overly uniform backgrounds). Existing preference metrics (e.g., ImageReward, PickScore) are relatively **insensitive** to such local artifacts but more sensitive to global semantics, causing early-stage models to sometimes achieve higher preference despite worse visual details.
>
> We consistently observe this trend across benchmarks and metrics; notably, **in the later stages of training, visual quality metrics remain relatively stable, while human preference metrics exhibit larger fluctuations**. Therefore, the observed preference drop reflects a misalignment between automatic preference metrics and perceptual quality, rather than true quality degradation.
>
> Compared to vanilla DMD, DP-DMD exhibits more **stable training dynamics** by preventing later-stage optimization from overriding early diversity signals, thus mitigating collapse. In practice, we determine stopping points using a **combination of metrics and visual inspection**, rather than a single preference score. For example, for SD3.5-M (2B), we find that approximately **20K** samples (3500 iterations) are sufficient to obtain strong visual quality. For larger-scale models, we expect that more training samples or iterations may be required to reach a similar convergence point.
>
> We thank the reviewer for raising this important question, and will add the above discussions in the revision.

---

> > ### Author Rebuttal · Reviewer_xLzh · 2026-04-03
> >
> > The rebuttal is detailed and clear.

---

> > > ### Author Response · Authors · 2026-04-03
> > >
> > > Many thanks! We sincerely appreciate Reviewer xLzh’s thoughtful comments, which have been very helpful in improving the paper. We will incorporate the additional analysis into the revised version.

---

### Official Review · Reviewer_8yyd · 2026-03-10

**Soundness:** 3
**Presentation:** 2
**Significance:** 2
**Originality:** 3
**Overall Recommendation:** 4
**Confidence:** 2

**Summary:**

The paper focuses on addressing the issue of sample diversity loss (mode collapse) caused by the use of reverse KL divergence during the distillation process of diffusion models. The core idea, based on the observation that early steps in the denoising process determine global structure/diversity while later steps handle details/quality, is to propose a role-separated distillation framework called DP-DMD.

**Compliance With Llm Reviewing Policy:**

Affirmed.

**Ethical Review Concerns:**

None.

**Final Justification:**

The authors have addressed my concerns.

**Key Questions For Authors:**

1.Can you provide a theoretical explanation for why applying the v-prediction loss specifically at the first step can systematically counteract the mode collapse induced by reverse KL divergence?
2.Can you provide an analysis of scenarios where the method might fail?
3.Can the definition and evaluation of diversity be made more explicit and multi-faceted?

**Limitations:**

Lack of Theoretical Analysis: Although the method is intuitive and effective, there is a lack of deeper theoretical analysis to prove why applying the v-prediction loss at the first step can systematically counteract the mode collapse caused by reverse KL divergence. The analysis is primarily based on experimental observations.

**Strengths And Weaknesses:**

Strengths:
1.Reasonable Motivation: It accurately identifies the core defect of distribution matching distillation methods like DMD, which is the mode collapse and reduced diversity caused by reverse KL divergence.
2.Simple and Efficient Method: By employing only two core operations—role separation and gradient stopping—it effectively mitigates the diversity reduction problem without introducing any additional networks (such as perceptual models or discriminators) or increasing computational overhead.
Weaknesses:
1.Limitations of Diversity Supervision: Confining diversity supervision entirely to the first step, while based on observations, may be an oversimplification. The paper acknowledges this in the Limitations section but does not provide quantitative analysis to illustrate under what circumstances this "one-size-fits-all" separation approach might fail.
2.Lax Experimental Comparison: Although the system-level comparison in Table 4 aims to demonstrate practical effectiveness, the authors also acknowledge differences across methods in terms of training data, CFG scale, optimization budget, etc. This makes the conclusion of this comparison ("achieving competitive open-source performance") somewhat vague and difficult to use as a strict performance ranking.
3.Limited Definition and Metric for "Diversity": The evaluation primarily relies on similarity based on DINO and CLIP features (indirectly reflecting diversity). While this is a common practice, diversity itself is a multi-dimensional concept (e.g., composition, color, texture).

---

> ### Author Rebuttal · Authors · 2026-03-27
>
> We sincerely appreciate this reviewer's insightful feedback and careful review of our manuscript. Please find what below our detailed point-to-point responses to all the comments of this reviewer. We hope our responses can address this reviewer's concerns.
>
> ---
>
> **`[W1 & Q2] Regarding Failure Scenarios`**
>
> We thank the reviewer for raising this insightful question. As noted in our limitation section (Lines 427–432), the assumption that diversity can be fully controlled in the first step may not always hold. A representative failure case is **composition and spatial reasoning**. For instance, in Figure 4 (second prompt), DP-DMD often produces a consistent layout (e.g., the cat appears on the left of the dog), indicating reduced compositional diversity. We attribute this to aggressively distilled few-step models, where object layout is not fully determined in the first step and is further refined in later steps. To quantitatively validate this, we generate 50 samples per prompt using DMD and DP-DMD, and measure the distribution of relative spatial positions.
>
> \begin{array}{lcc}
> \hline
> \text{Method} &\text{cat on the left} &\text{cat on the right}\newline
> \hline
> \text{DMD} &41 &9\newline
> \text{DP-DMD} &38 &12\newline
> \hline
> \end{array}
>
> As shown in the above table, spatial distributions remain similar between DMD and DP-DMD, indicating limited improvement in compositional diversity (e.g., left-right arrangement). However, we observe that DP-DMD exhibits richer diversity in other aspects, such as color, animal breeds, and overall scene composition, indicating that its gains extend beyond coarse spatial layouts. Overall, while being limited in attributes (e.g., identity or composition) that are partially determined in **later steps**, DP-DMD achieves substantially improved overall diversity over DMD with a simple and efficient design.
>
> ---
>
> **`[W2] Regarding Experimental Comparison`**
>
> We appreciate the reviewer’s thoughtful comments. We would like to clarify that the primary goal of this work is **NOT** to establish a strict performance ranking against existing open-source models, but to address a key limitation of DMD-based distillation, namely the loss of **sample diversity**, and to demonstrate the effectiveness of our method in this context. Accordingly, we consider the controlled comparisons in Table 3, together with the qualitative results in the Appendix. **All variants share the same backbone, training setup, and inference budget**, enabling a fair and rigorous evaluation.
>
> The purpose of Table 4 is to provide a system-level reference. Despite its simplicity, our method achieves performance comparable to existing open-source approaches, showing that a lightweight, role-separated loss can maintain competitiveness while improving diversity, which we believe is a meaningful takeaway.
>
> ---
>
> **`[W3 & Q3] Regarding Diversity Metric`**
>
> We thank the reviewer for the insightful comment. We agree that diversity is inherently multi-dimensional (e.g., composition, structure, color, texture, semantics), and there is **currently not a universally accepted metric** that can fully capture all these aspects. Designing such a metric would require jointly modeling multiple factors across feature spaces, which remains an open challenge.
>
> Given this, we follow established practice in prior work and adopt DINO and CLIP feature similarities as complementary proxies for diversity, enabling fair comparison with existing DMD-based methods. Our goal is not to propose a new diversity metric, but to evaluate diversity under a standard protocol. To further provide more direct and multi-faceted evidence, we additionally include a human user study (Appendix Sec. D, Fig. B), as **human evaluation remains the most reliable way** to assess diversity in generative models.
>
> ---
>
> **`[Q1 & L1] Regarding The Theoretical Analysis`**
>
> Thanks for pointing this out. In this work, we provide a more principled view based on the flow-matching formulation (Eq. 2). The v-prediction objective can be interpreted as learning a **transport map** from the noise distribution to the target distribution by matching the conditional velocity field. In particular, the optimal velocity field in Eq. 2 is defined with respect to the teacher-induced intermediate distribution, thus encouraging probability mass to **spread over the target support rather than concentrate on a few high-density regions**.
>
> When we apply this constraint at the first step (Eq. 6), we effectively anchor the student’s initial transport direction to this teacher-aligned velocity field. As a result, each noise sample is guided toward the correct region of the target distribution according to the teacher’s conditional transport, which enforces a form of support-level alignment between the student and teacher. This acts as a **distribution-level regularization** that counterbalances the mode-seeking bias of reverse KL, leading to improved coverage and diversity.

---

> > ### Author Rebuttal · Reviewer_8yyd · 2026-04-04
> >
> > The authors have addressed my concerns, and I will raise the score.

---

> > > ### Author Response · Authors · 2026-04-04
> > >
> > > Many thanks! We sincerely appreciate Reviewer 8yyd’s insightful comments, which have been very helpful in improving the paper. We will incorporate the additional experimental results and analysis into the revised version.

---

### Official Review · Reviewer_V512 · 2026-03-11

**Soundness:** 3
**Presentation:** 3
**Significance:** 3
**Originality:** 3
**Overall Recommendation:** 4
**Confidence:** 4

**Summary:**

This paper propose DP-DMD (Diversity-Preserved DMD), a role-separated distillation framework in which the first distilled step is supervised by a flow-matching target-prediction objective, while the remaining N−1 steps are trained under the standard DMD loss. The motivation is grounded in an empirical observation that early denoising steps determine global structure and diversity, while later steps refine perceptual quality.

**Compliance With Llm Reviewing Policy:**

Affirmed.

**Final Justification:**

My concerns have been addressed.

**Key Questions For Authors:**

see weaknesses

**Limitations:**

yes

**Strengths And Weaknesses:**

Strengths:
1. Well-structured and clear: The paper is well-written and logically organized. The motivation, background, method, and experiments flow naturally, making the manuscript easy to follow.
2. Simple yet effective method: The proposed approach is effective without relying on a perceptual backbone, discriminator, auxiliary networks, or additional ground-truth images. By performing all operations within the latent space, the pipeline remains compact, stable, and memory-efficient.
3. Effective Mitigation of Mode Collapse: The paper addresses the diversity degradation issue inherent in standard DMD. The results validate that DP-DMD recovers the diversity lost in vanilla DMD, ensuring varied outputs while preserving the high visual quality associated with GAN and LPIPS variants.

Weaknesses:
1. Missing baselines and analysis on diversity: Table 4 lacks the results for Hyper-SD (4 NFE). Furthermore, there is insufficient analysis regarding why DP-DMD exhibits inferior diversity compared to the 8 NFE setting. This analysis need not be limited to training data or hyperparameters. For instance, Hyper-SD belongs to the family of Consistency Models (CMs). Since DP-DMD supervises the first denoising step combined with the DMD loss, it can be viewed as a hybrid of distribution matching distillation and consistency distillation. It remains unclear whether the diversity improvement in DP-DMD stems from the introduction of this consistency-like distillation. Have the authors conducted consistency distillation experiments on SDXL to explore how it affects the diversity capabilities of the distilled model? Does it effectively preserve the model's generative diversity?
2. Impact of Teacher CFG on diversity: During the training of DP-DMD, is there a significant difference in the final diversity improvement when the real teacher utilizes CFG(cfg_scale>1) versus when it does not (cfg_scale=1)? I would expect to see an ablation study or analysis demonstrating how the diversity capabilities of DP-DMD evolve when trained under different teacher CFG parameter configurations.

---

> ### Author Rebuttal · Authors · 2026-03-26
>
> We sincerely thank this reviewer for the insightful feedback and the thorough evaluation of our manuscript. Please find what below our detailed point-to-point responses to all the comments of this reviewer. We hope our responses can address this reviewer's concerns.
>
> ---
>
> **`[W1-1] Regarding Hyper-SD (4 NFE) Results`**
>
> Thanks for pointing this out. We have added the results of Hyper-SD (4 NFE), as shown in the table below.
>
> \begin{array}{lcccccc}
> \hline
> \text{Method} &\text{DINO$\uparrow$} &\text{CLIP$\uparrow$} &\text{VQ-R1$\uparrow$} &\text{MIQA$\uparrow$} &\text{ImgR$\uparrow$} &\text{PicS$\uparrow$}\newline
> \hline
> \text{\emph{Pick-a-Pic}}\newline
> \text{Hyper-SD} &0.202 &0.198 &4.302 &0.922 &0.830 &20.75\newline
> \text{DP-DMD} &0.162 &0.181 &4.673 &1.001 &1.128 &21.15\newline
> \hline
> \text{\emph{COCO-10K}}\newline
> \text{Hyper-SD} &0.295 &0.231 &3.920 &0.913 &0.653 &21.52\newline
> \text{DP-DMD} &0.197 &0.174 &4.672 &1.048 &1.034 &22.29\newline
> \hline
> \end{array}
>
> We see that although Hyper-SD maintains diversity, it does so at a **substantial price of generation quality**, with notably worse visual fidelity and human preference, underperforming other open-source baselines across multiple quality metrics (Table 4). In contrast, we argue that **few-step generation should enhance diversity without incurring such a pronounced loss in quality (relative to the base multi-step model)**, which is the key strength of our method.
>
> ---
>
> **`[W1-2] Regarding the Analysis of Consistency-like Distillation`**
>
> We sincerely thank this reviewer for this insightful comment. We **FULLY** agree that the diversity improvement in DP-DMD is related to the introduction of a consistency-like supervision signal, motivating our design (Sec. 4.1). Generally speaking, improving sample diversity through perceptual loss, adversarial loss, or consistency distillation can be viewed as guiding the student distribution toward the data or teacher distribution, as they all rely on introducing **external information** to regularize the learning process. Therefore, introducing consistency distillation on top of DMD, which has a strong ability to preserve visual quality under extremely limited NFEs, is both reasonable and effective (following the **“quality-first, then diversity” principle**). The key lies in how such supervision is incorporated in a **simple and effective manner**.
>
> For the **SDXL** setting, we have already included consistency distillation experiments in the main paper (Table 3), where introducing a consistency-like objective on top of DMD, unlike a naive CM+DMD combination, DP-DMD enforces role separation with first-step diversity supervision and gradient blocking, leading to a clear improvement in diversity (DINO: 0.109 to 0.173 on Pick-a-Pic, 0.139 to 0.204 on COCO), confirming its effectiveness in alleviating diversity collapse while maintaining strong visual quality.
>
> Overall, while consistency-like distillation can improve diversity, DP-DMD leverages it effectively to enhance diversity without sacrificing visual quality. We will include this analysis in the revised manuscript.
>
> ---
>
> **`[W2] Regarding the Ablation Study on CFG`**
>
> Thanks for the constructive suggestion. We **AGREE** that teacher CFG is crucial for diversity in diffusion distillation, and we have conducted additional ablations to analyze its impact within DP-DMD on the Pick-a-Pic dataset.
>
> \begin{array}{lcccccc}
> \hline
> \text{CFG} &\text{DINO$\uparrow$} &\text{CLIP$\uparrow$} &\text{VQ-R1$\uparrow$} &\text{MIQA$\uparrow$} &\text{ImgR$\uparrow$} &\text{PicS$\uparrow$}\newline
> \hline
> \text{SD3.5-M (CFG=1.0, 60 NFEs)} &0.457 &0.357  &3.869 &0.932 &-0.317 &20.38\newline
> \text{DMD (CFG=3.5)} &0.137 &0.133 &4.649 &1.016 &1.189 &21.75\newline
> \hline
> \text{1.0 (no CFG)} &0.477 &0.335 &2.788 &0.888 &-0.402 &20.13\newline
> \text{3.5} &0.179 &0.182 &4.646 &1.017 &1.142 &21.76\newline
> \text{6} &0.192 &0.175 &4.666 &1.010 &1.206 &21.61\newline
> \text{8} &0.172 &0.205 &4.628 &1.028 &1.260 &21.75\newline
> \text{10} &0.166 &0.146 &4.656 &1.021 &1.293 &21.69\newline
> \hline
> \end{array}
>
> As shown in the table, there is a **clear difference** between using CFG and without using CFG (cfg=1). While it is known that CFG can significantly improve generation quality at the cost of reduced diversity, the teacher with cfg=1 produces noticeably lower-quality samples. In step distillation, the teacher’s generation quality directly determines the upper bound of the student’s performance. Our experiments reveal two key findings:
>
> - Increasing CFG leads to improved visual quality, but tends to reduce diversity, which aligns with the commonly observed quality-diversity trade-off in literature.
> - Importantly, under DP-DMD, even at relatively high CFG values, the student model **maintains strong diversity** compared to the DMD baseline at comparable quality levels.
>
> We will include these results and analysis in the revised manuscript.

---

> > ### Author Rebuttal · Reviewer_V512 · 2026-04-03
> >
> > Both weaknesses have been substantively addressed with new experimental results.
> >
> > W1-1: The added Hyper-SD (4 NFE) results show a clear quality-diversity trade-off that DP-DMD avoids — a direct and convincing response.
> >
> > W1-2: The authors' agreement that diversity improvement is related to consistency-like supervision is appreciated. The existing Table 3 results (DINO: 0.109→0.173 on Pick-a-Pic, 0.139→0.204 on COCO) with role separation and gradient blocking already demonstrate this empirically in the main paper. The concern is resolved.
> >
> > W2: The CFG ablation results are clear and interpretable. DP-DMD maintaining strong diversity across CFG values relative to the DMD baseline is a meaningful finding that should be included in the revised paper.

---

> > > ### Author Response · Authors · 2026-04-04
> > >
> > > Many thanks! We sincerely appreciate Reviewer V512’s thoughtful comments, which is very valuable for improving the paper. We will incorporate the additional experimental results and analysis into the revised version.

---

### Official Review · Reviewer_Ehje · 2026-03-11

**Soundness:** 3
**Presentation:** 3
**Significance:** 3
**Originality:** 3
**Overall Recommendation:** 5
**Confidence:** 3

**Summary:**

The paper proposes a method for fast image synthesis using diffusion, while preserving output diversity. This is achieved through extending distribution matching distillation (DMD) to be diversity preserving. The proposed model can run in 4 NFEs (number of function evaluations) to produce high quality imagery compared to base model that runs at 60 NFEs while maintaining more output diversity than competing methods. This is done through separation of diffusion supervision into diversity and quality stages in a unified manner. This is supported by a set of experiments demonstrating that it can better preserve output diversity compared to regular distillation methods.

**Compliance With Llm Reviewing Policy:**

Affirmed.

**Key Questions For Authors:**

It would be great to see performance comparisons of different systems, currently the only comparison is number of NFEs but not the total inference time etc.

**Limitations:**

Yes

**Strengths And Weaknesses:**

Strengths:
- The paper is well justified and explained, it is clear to read and follow
- The topic is interesting and worthwhile, distillation is known to collapse the output diversity and this method demonstrates a way to tackle that. While diversity is still worse compared to a base model it is better than other baselines (DMD, DMD-LPIPS, DMD-GAN)
- The paper clearly describes the limitations of approach and trade-off between quality and diversity
- The proposed approach is conceptually simple and does not add complications to inference pipeline

Weaknesses:
- While more diversity is preserved it is still somewhat behind the base models from which distillation happens, further there is a drop in quality metrics as well (Table 3)
- It would be great to see performance comparisons of different systems, currently the only comparison is number of NFEs but not the total inference time etc.
- Trivia: In the example Figure 5 DP-DMD example makes Valerian a woman, which is incorrect, while SD3-M outputs it correctly

---

> ### Author Rebuttal · Authors · 2026-03-25
>
> We are truly grateful for this reviewer's valuable insights and careful evaluation on our manuscript. Please find what below our detailed point-to-point responses to all the comments. We hope our responses can address this reviewer's concerns.
>
> ---
>
> **`[W1] Regarding the Reduction in Diversity and Quality Compared to the Base Model`**
>
> We thank this reviewer for this insightful comments. In diffusion model distillation, distilling a multi-step teacher model into a few-step student (e.g., 4 steps) while maintaining both quality and diversity is inherently **challenging**. Extensive prior research and industry practice suggest that DMD and related approaches are currently effective solutions for few-step distillation in maintaining generation quality, while their main limitation lies in a noticeable reduction in sample diversity.
>
> It is important to note that the base models (e.g., SD3.5-M or SDXL) operate with 30+ sampling steps, allowing for **more extensive exploration** of the data distribution and naturally resulting in higher diversity and strong quality. This behavior serves as an **upper bound**, rather than a limitation of our method. As shown in Table 3 , although slight degradation in certain quality metrics can be observed compared to the teacher model, our DP-DMD consistently achieves competitive visual quality and, in some settings (e.g., SDXL backbone), even improves human preference scores over both the base DMD baseline and the original model. More importantly, DP-DMD significantly **improves sample diversity while maintaining comparable generation quality**, which directly addresses the core challenge of mode collapse in DMD-based distillation and constitutes the main contribution of our work.
>
> ---
>
> **`[W2 & Q1] Regarding Comparison with Other Systems`**
>
> We sincerely thank this reviewer for the suggestion. We would like to clarify that our method does not modify the model architecture, but instead focuses on **optimization**, similar to mainstream open-source distillation methods (see Table 4). Therefore, the per-step computational cost remains comparable, and NFEs serve as a standard proxy for efficiency.
>
> To provide a more comprehensive evaluation, we have added additional experiments comparing the **actual inference time** of our method with the base model. Specifically, we report the total inference time under different settings, including batch sizes of 1, 4, 8, 16, and 32, at a resolution of $1024 \times 1024$, on a single 80GB NVIDIA A800 GPU. The results are summarized in the table below.
>
> \begin{array}{l|c|c|c|c|c}
> \hline
> \text{Method} & \text{BS=1} &  \text{BS=4} &  \text{BS=8} &  \text{BS=16} &  \text{BS=32}  & \text{NFEs} \newline
> \hline
> \text{SD3.5-M (base model)} &6.116\text{s}   &22.465\text{s}   &44.230\text{s}    &87.893\text{s}    &177.926\text{s}   & 60 \newline
> \text{DP-DMD} &0.658\text{s} &2.198\text{s} &4.214\text{s} &8.200\text{s} &18.189\text{s}         & 4 \newline
> \hline
> \end{array}
>
> As shown in the table above, our method achieves consistent $9\times$ speedup in total inference time across all batch sizes, which directly aligns with the reduction in NFEs (60 to 4). These results will be included in the revised version of the paper.
>
> ---
>
> **`[W3] Regarding the Incorrect Example in Figure 5`**
>
> Thanks for pointing out this issue. To avoid potential confusion, we will revise Figure 5 in the revised version to include additional samples. We appreciate the reviewer for bringing attention to this issue.

---

> > ### Author Rebuttal · Reviewer_Ehje · 2026-04-01
> >
> > Rebuttal clarified efficiency of algorithm.

---

> > > ### Author Response · Authors · 2026-04-02
> > >
> > > Many thanks! We sincerely appreciate Reviewer Ehje’s thoughtful comments, which have been very helpful in improving the paper. We will incorporate the additional experimental results and analysis into the revised version.

---

### Decision · Program_Chairs · 2026-04-30

**Decision:**

Accept (regular)

**Comment:**

Four knowledgeable reviewers went over this submission. Their main concerns may be summarized as:
1. Diversity-quality trade-off (Ehje, 8yyd, V512):
   - The method does not recover fully the base model diversity.
   - Diversity supervision is only applied to the first step, which may be an oversimplification.
   - Insufficient validation of diversity: missing baselines and analyses to understand where diversity gains stem from.
    - Missing ablation to understand how teacher guidance scale affects the quality-diversity.
    - Limited diversity metrics.
2. Lack of inference time comparison: missing wall-clock times (Ehje).
3. Lack of theoretical justification for some steps (8yyd).
4. Unclear hyper-parameter sensitivity and selection (xLzh).
5. Training dynamics: unclear why preference decreases with longer trainings and how to know when to stop the training (xLzh).

The rebuttal addressed the authors' concerns by arguing the base model should be considered an upper bound, providing quantitative comparisons and acknowledging limitations. The authors discussed missing baselines and agreed that diversity improvements relate to consistency-like supervision. The rebuttal also included a CFG ablation showing that the method maintains strong diversity at higher CFG values relative to the baseline, and an inference-time table across batch sizes showing the speedup achieved. To address the theoretical justification concern, the rebuttal provides an interpretation that appears rather intuitive and is accepted by the reviewer. The rebuttal also covers hyper-parameter sensitivity/choices and discusses training dynamics, attributing some observations to reverse-KL mode collapse.

After rebuttal, all four reviewers mark their concerns as fully resolved and they unanimously recommend acceptance. The AC agrees with the reviewers and recommends to accept.